# GaneStat—A comprehensive design and modular analysis of portable, low-cost, and high-accuracy potentiostat

Isa Anshori[1,2,3¤☯*], Infall Syafalni[4,5☯*], Christian Reivan[4☯*],
Iqbal Fawwaz Ramadhan[4], Theodore Maximillan Jonathan[4], Rizky Indah Sari[4],
Uperianti[6], Trio Adiono[4,5], Chih-Yu Chang[7], Akhmadi Surawijaya[4,5]

**1** Lab on Chip Laboratory, Biomedical Engineering Department, School of Electrical Engineering and Informatics, Institut Teknologi Bandung, Bandung, Indonesia, **2** Research Center for Nanoscience and Nanotechnology, Institut Teknologi Bandung, Bandung, Indonesia, **3** ITB Center for Health snd Sports Technology (PTKK-ITB), Institut Teknologi Bandung, Bandung, Indonesia, **4** School of Electrical Engineering and Informatics, Institut Teknologi Bandung, Bandung, Indonesia, **5** University Center of Excellence on Microelectronics, Institut Teknologi Bandung, Bandung, Indonesia, **6** Department of Nanotechnology, Graduate School, Institut Teknologi Bandung, Bandung, Indonesia, **7** Material Science and Engineering Department, National Taiwan University of Science and Technology, Taipei, Taiwan

¤ Current address: Biomedical Engineering Department, School of Electrical Engineering and Informatics, Institut Teknologi Bandung, Bandung, Indonesia
☯ These authors are contributed equally to this work.
* isaa@staff.stei.itb.ac.id (IA); infall@ieee.org (IS); christianreivan@gmail.com (CR)

## Abstract

Electrochemical research has been developing with the advancement of laboratory equipment and sophisticated technologies. One of which is the portable potentiostat, which has been utilized to analyze various samples and help characterize their electrochemical properties. In this paper, we propose a comprehensive design and modular analysis of our proposed potentiostat called GaneStat. The proposed potentiostat has a low cost, portable, and high accuracy with battery-powered and electrical overstress circuit protection. The proposed analog front end of the potentiostat consists of several modules such as unipolar-to-bipolar converter (UBC), buffer, current-to-voltage converter (CVC), bipolar-to-unipolar converter (BUC), two-stage sallen key low-pass filter, and ADC input protection unit (AIPC). A maximum wide dynamic range of 98.95 dB provides flexibility in terms of current measurement. The potentiostat is also equipped with a dedicated power management circuit and an overstress protection circuit. Multi mode measurements are provided in our system for Cyclic voltammetry (CV), differential pulse voltammetry (DPV), Linear Sweep Voltammetry (LSV), and Chronoamperometry (CA) experiments with preliminary test on redox probe solution to validate the performance of the device. The final design occupies relatively small space of 6.95cm × 6.85 cm × 3.26 cm with 83.1 gr of weight, including the battery and the case. The potentiostat operates with sweep voltage within ±2.5 V with a 1.2 mV resolution, and it can measure current from 10 nA to 10 mA with 0.53 nA and 0.3 $\mu$A resolutions, respectively. The potentiostat costs only $98.55

**Data availability statement:** All relevant data are within the manuscript.

**Funding:** This work is supported by Direktorat Penelitian dan Pengabdian kepada Masyarakat (DPPM), Direktorat Jenderal Riset dan Pengembangan, Kementerian Pendidikan Tinggi, Sains, dan Teknologi (Kemendiktisaintek) Republik Indonesia 2025 through Skema Penelitian Dasar program under grant no. 364/IT1.B07.1/SPP-DRI/V/2025. The authors would also like to acknowledge funding from Bandung Institute of Technology through the Riset Unggulan 3P Tahun 2024 program [Grant No. 766.15/IT1.B07.5/TA.00/2024]. The funders had no role in study design, data collection and analysis, decision to publish, or preparation of the manuscript.

**Competing interests:** The authors have declared that no competing interests exist.

for the prototyping. This work is useful for laboratory applications in chemical, pharmacy and medical industries.

## Introduction

Electrochemical analysis are quantitative techniques that involves the application of electrical signals to determine the electrochemical properties of a chemical solution. There is a wide range of chemicals that can be analyzed, including biological compounds. The application of electrochemical analysis to determine the presence of certain chemicals typically involves a set of electrode transducers that connected to a potentiostat instrument to translate the chemical signals into electrical signals which is digitally easier to process. The potentiostat has permitted various innovations exclusively in sensor development and materials science by offering precise control of electrical potentials and current flow [1]. In addition, the requirement for flexibility and portability has increased specific demand toward potentiostats development. This is because traditional potentiostats are usually bulky and require a cable for the power source, limiting their mobility. Thus, the introduction of portable potentiostats has transformed the way tests are carried out, providing unparalleled ease and broadening the boundaries of scientific experiments. Portable potentiostats in the form of small, light, wireless, and battery-powered, allow researchers to take the laboratory to the field, point-of-care settings, or distant sites [2,3,5].

Generally, there are two approaches for achieving portable potentiostat, either utilizing a dedicated chip (on-chip system) that has already included a built-in potentiostat circuit or implementing the analog front-end (AFE) using discrete components (off-chip system). A miniaturized potentiostat has been proposed in [6] where LMP91000 was employed as the AFE. Although a small form factor was obtained, the voltage range it can span is limited. Also, it lacks of customizability as it is impossible to modify LMP91000 itself unless fabricating a new chip. Nevertheless, on-chip approach for sensor reader development has clear advantages in many aspects, such as miniaturization (portability and integration), power efficiency, performance (precision and fast response time), scalability (parallel measurements and customization), connectivity, and robustness (noise reduction and durability). These aspects show the role of on-chip system as a common standard in commercial potentiostat. The second approach uses discrete components integrated into a PCB to reach portability like the one which is presented in [7]. In the work, Arduino-based potentiostat is developed for teaching electrochemistry fundamentals and applications. The work in [8] also used discrete components to build a potentiostat where a four-channel front-end interface is proposed with programmable and wearable solutions to sense different analytes using several electrochemical methods. Another example is the portable potentiostat presented in [9]. It consumed a total size of 10.5 cm × 5.8 cm × 2.5 cm and a weight of 41 g. However, the power supply as well as the data transfer between the potentiostat and the software still utilize a USB, resulting in low flexibility. The proposed design in [10] obtains the power supply from 3 lithium-ion

batteries. However, a wired connection is still necessary to provide the communication path between the potentiostat and a PC. Furthermore, the potentiostat is equipped with a touchscreen display which makes it easier to visualize the results but consumes a lot of space.

Apart from the portability requirement, the promising potentiostat should also possess robust accuracy and a wide current range while maintaining low-cost aspects at the same time. The accuracy is evident from the applied voltage and the measured current resolutions. It is desired to have the applied voltage resolution of less than 2-3 mV for better performance [11] and this requirement is satisfied in [2,12–15]. The applied voltage resolution in [12] is 1.6 mV with a voltage span of 6.6 V, in comparison to 0.7 mV and 3 V for the work in [2]. The proposed design in [12] can measure current up to 13.75 mA, while it is only close to 500 $\mu$A for the work in [2]. However, the work presented in [2] is much superior in terms of the current resolution, which is 0.24 $\mu$A, or, about 4 times smaller than that in [12]. Both previous designs had only one range for current reading, resulting in low flexibility for wider measurement ranges. Potentiostats with multi range features were found in [13–15] where the measurement ranges are 1 mA to 20 mA (6 options), 0.25 mA to 250 mA (4 options), and 200 nA to 20 mA (6 options), respectively. Another portable potentiostat with a development cost of around \$30 was found in [16]. Despite being sufficiently low in cost, it can only deliver a voltage resolution of 15.7 mV and sense a maximum current of 200 $\mu$A, thus hurting its performance. For the same reason, a lower production cost of only \$21.4 was obtained in [2], but the trade-offs include having a narrower applied voltage range than that in [16] and a large area. Putting all this together, we can see that there is a compromise between portability and performance, not to mention the space consumed by a battery if employed as the power source for the potentiostat. Some recent researches on specific IC for potentiostat has been conducted in [17,18]. The work in [17] presented a potentiostat with 96 channels implemented in 40 nm CMOS technology. The proposed method implemented an on-chip flexible digital feedback controller. Moreover, the work in [18] reduces the use of amplifiers and bulky passive components by a digital control architecture. The work is implemented in 180 nm CMOS process and occupies only 0.266 $mm^2$. These works optimize the potentiostat with custom digital controller implemented in IC technology. However, in practice, an IC fabrication requires high cost and mass productions to meet revenues in the industries. It is still an open opportunity to produce a low-cost device with better performances, although the best performances are from the IC solutions.

In this paper, we present GaneStat as shown in Fig 1, a comprehensive design and analysis of modular potentiostat for low-cost and high-accuracy electrochemical device. We selected the system reference from commercial on-chip potensiostat module EmStat4M to evaluate the performance of our off-chip design. Some targeted improvement of performance, such as voltage range, current range and current resolution are of our concern. This work shows a comprehensive design and analysis of a portable potentiostat. The mathematical analysis of its analog front-end components is explained in details. The components are the unipolar to bipolar converter, the current to voltage converter, the bipolar to unipolar converter, the two-stage Sallen-key low-pass filter (LPF), and the analog to digital converter (ADC) input protection. We have also taken into account the electrical overstress and load-sharing problems that might be present by designing dedicated circuits for the circuit protection and power path controller. The proposed electrical overstress protection was used to limits the voltage for the ADC input.

## Materials and methods

### Chemicals, sensor, and instrument

Potassium hexacyanoferrate (K$_3$[$Fe(CN)_6$]) was purchased from Loba Chemie (Mumbai, India). To evaluate the device's performance, cyclic voltammetry is applied to the K$_3$[$Fe(CN)_6$]) solution at concentrations of 2.5, 5, 10, 15, and 20 mM in a solution of 10 mM phosphate-buffered saline (PBS), pH 7.4. The electrochemical sensors used were the unmodified SPCE from Zimmer & Peacock (Coventry, UK). In addition, sulfuric acid (H$_2SO_4$) was employed for the SPCE activation procedure as mentioned in [19]. Lastly, the Palmsens EmStat4M (Houten, Netherlands) module was utilized for performance validation and accuracy comparison of the proposed potentiostat.

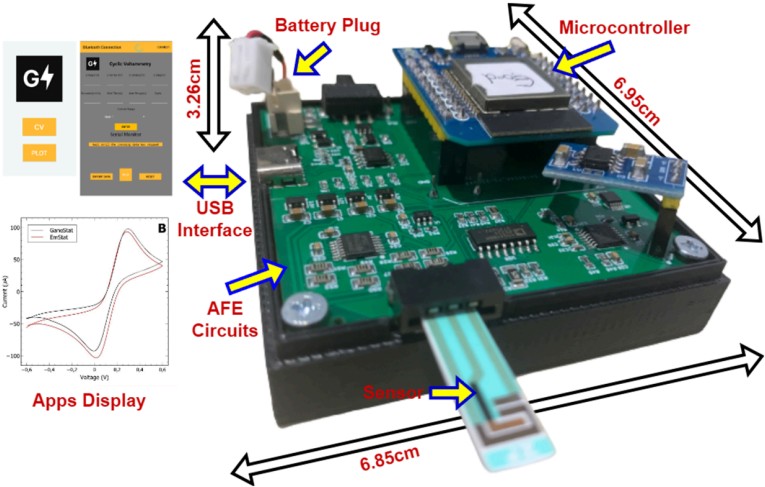

**Fig 1**. Overall illustration of the proposed GaneStat.

## Experiment method and system overview

Cyclic voltammetry (CV) analysis induces a cyclic and linear potential sweep at a certain rate (scan rate). The potential is swept negatively from the starting potential $E_1$ to the switching potential $E_2$„ which is referred to as the cathodic trace, and then followed by a sweep in the opposite direction, which is referred to as the anodic trace [20]. The potential sweep may be done for more than one cycle. As a consequence, some amount of electrical current can flow through the chemical in the form of electron transfer. The technique enables the measurement of oxidation and reduction processes, in addition to the identification of diverse oxidation states and reduction states in distinct chemical reactions [21]. The current response from this analysis also enables further study regarding the kinetics of the electron transfer process in redox reactions. While conducting CV analysis, one must use the appropriate scan rate, which is the rate of change of the cyclic potential sweep. When the potential changes too steeply over time, the oxidation/reduction reaction may never reach stability; hence, accurate results may not be obtained.

GaneStat comprises two systems, namely hardware and a graphical user interface (GUI), as seen in Fig 2. The hardware is made up of electrical components onboard that form together to perform signal processing and management, as will be further discussed later (Fig 2a). On the other hand, the GUI is an interface using which we interact with the hardware (Fig 2b). This interaction involves filling in analysis parameters, exporting data to CSV, and plotting data. The analysis parameters, such as initial and final sweep voltage (V), step size (mV), scan rate (mV/s), wait time (s), auto range duration (s), and the number of cycles, are arranged sequentially in a string in the background before being sent to the hardware. On the hardware side, the sequence is then parsed in such a way that the main controller within the hardware gets the information correctly and passes instructions to each component to act correspondingly. The output signal that has undergone several processes by the hardware is then sent back to the GUI in a string. Now the string consists of the current being measured and its corresponding sweep voltage, separated in between with a comma. Each current-voltage pair that is received by the GUI will be printed on a built-in serial monitor and appended to a storage one at a time. After all pairs have been completely sent from the hardware, the serial monitor will stop printing the result and begin saving the measurements into a CSV file only if we click the "export data" button. If so, the measurement data will be saved in local storage. We can plot the data by hitting the "plot" button. There will be a prompt to enter the desired CSV file, which will be displayed. To obtain a more in-depth analysis, such as getting the anode and cathode currents or voltages, we can export the CSV file to commercial graphing software. The proposed hardware design can be functionally broken down into

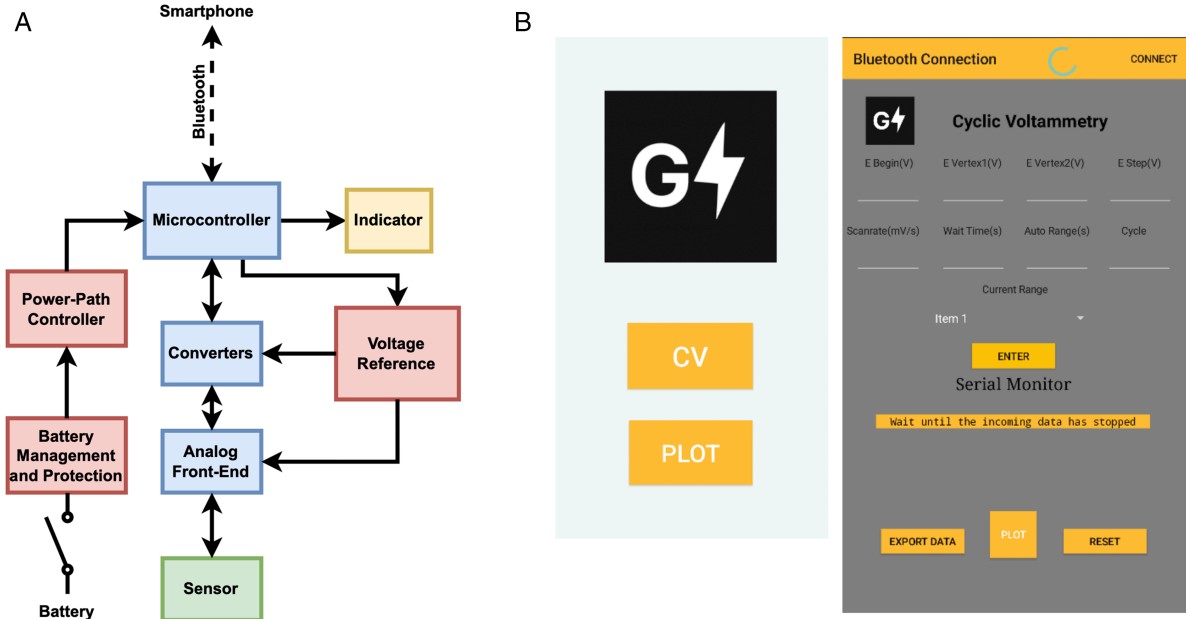

**Fig 2**. System Overview of GaneStat. (a) Functional Block Diagram of GaneStat. (b) GUI View in Smartphone.

several blocks: 1) Microcontrollers & Communication, 2) DAC & ADC, 3) Voltage Reference, 4) Battery Management & Protection System, Power-path Controller, and Load Sharing Concept, and 5) Analog Front-End.

## Hardware design

The proposed hardware design can be functionally broken down into several blocks, as illustrated in Fig 2. In general, two tasks that are handled by the hardware: signal processing and power supply management. The former is performed by microcontrollers, converters, Analog Front-End (AFE), indicators, voltage references, and step-up blocks, whereas the latter is constructed by battery management and protection for the system as well as power-path controller blocks.

**Microcontroller and communication.** The microcontroller is vital to the whole design as electrochemical analysis algorithms are implemented on it. The microcontroller also serves to control the signal flow. It maintains communication between hardware and software and helps interfacing peripherals, such as converters, indicators, and power-path controller blocks. In this design, the ESP32 D1 MINI is used. ESP32 is well known as the most widespread IoT System-On-Chip in the world [22] with the support of various features such as GPIOs, dual-core CPUs, and communications such as 802.11 b/g/n WiFi and Bluetooth. This module is run on a 32-bit, dual-core processor that can operate up to 240 MHz frequency. Furthermore, the ESP32 D1 MINI has an integrated onboard Bluetooth module. Thus, two functional modules are included in only one component. It is advantageous to keep the footprint as small as possible to achieve portability. Additionally, the ESP32 D1 MINI was selected since it can be programmed in the Arduino IDE, which has an intuitive user interface that can be learned by beginner to intermediate programmers with ease.

The potentiostat is designed to be able to communicate with Android devices. The data exchange is supported by a seamless Bluetooth 4.2 BR/EDR connection. Bluetooth Classic (BC) was chosen owing to its high data rate, which is beneficial for electrochemical analysis purposes. Although Bluetooth Low Energy (BLE) provides longer battery life, the proposed device can work up to a couple of hours before it has to be recharged, which is worthwhile. Besides, GaneStat provides another option for powering it up in case the battery runs out, i.e., a 5 V supply from a USB Type-C.

**DAC and ADC.** A digital-to-analog converter (DAC) is used to convert the digital code representation of the signal to its analog form. The DAC used in this design was MCP4725. It has several features, including 12-bit resolution, buffered output, and an interface involving only two wires using the I2C protocol. DAC is related to the applied voltage's resolution, and it limits the maximum scan rate of the potentiostat. In this design, the reference voltage for the DAC module is 3.3 V, and consequently, the least-significant bit (LSB) size or the applied voltage's resolution, $\Delta_{DAC}$, can be calculated as

$$\Delta_{DAC} = \frac{3.3}{2^{12}} = 0.8 \; mV. \tag{1}$$

The microcontroller itself can only process digital information. Thus, the electrochemical result, which is inherently analog, needs to be converted back to its digital representation. This process of converting an analog signal into its digital form employs the use of an analog-to-digital converter (ADC). In this design, ADS1118 [23] was used as the ADC, which offers up to a 16-bit resolution, wide input range, low power consumption, and is equipped with an SPI interface. SPI, other than the I2C protocol, was opted for the ADC because it is multiple times faster than I2C [24]. Besides, the ESP32 D1 MINI only suffices for one I2C communication, where the option left is to find an ADC that adopts the SPI protocol for the communication interface. ADS1118 is based on delta-sigma architecture, and it was chosen considering that its resolution is much higher than that of SAR architecture because of its noise-shaping feature. Quality, other than sampling rate, is more considered because, when dealing with electrochemical reactions, every signal point that fails to be sampled can potentially damage the result. In other words, the ADC is related to the accuracy of the measurement of the device. With a 16-bit resolution and a 3.3 V reference voltage, one LSB size exhibited by the ADC, $\Delta_{ADC}$, is calculated as

$$\Delta_{ADC} = \frac{3.3}{2^{16}} = 50 \; \mu V. \tag{2}$$

**Voltage reference.** To power the microcontroller, we passed the output from the battery to the MT3608, a step-up converter, to ensure the incoming supply to the ESP32 was adequate. In addition, we utilized SPX3819M5, a low-noise low-dropout (LDO) voltage regulator, to provide stable and regulated 1.2 V, 2.5 V, 3.3 V, and 5 V supplies to the converters (ADC and DAC) and the AFE. As the LDO regulator requires headroom to work properly, we also used another MT3608 to raise 3.3 V coming from the ESP's 3.3 V pin to a few volts above 5 V before being sent to the LDO regulator. This is to accommodate all four output voltages. The op-amps included in the AFE will be powered by a $\pm 5$ V supply. However, we only have positive voltage references so far. Therefore, we also employed LM2662, a switched-capacitor positive-to-negative voltage converter, to obtain the –5 V supply.

**Battery management and protection system, power-path controller, and load sharing concept.** Generally, the proposed potentiostat is designed to be powered by a rechargeable LiPo battery, and thus having a battery management and protection unit is indispensable to managing charging as well as discharging processes. The system was implemented using TP4056. We also availed of a battery protection module and an NMOS to protect the battery from overcharge current and short circuit current. To charge the battery, it is necessary to plug in an external 5 V power supply via a USB-C charging cable. The charging-discharging cycle of the LiPo battery is characterized by constant-current and constant-voltage curves. The process is initiated when TP4056 injects the battery with a constant, predetermined current, thereby increasing the voltage across the battery until it reaches its nominal value and stays at the same position afterward, as the name suggests. During the constant-voltage period, the current decays and eventually terminates whenever it falls to 10% of the initial value.

To make the system more efficient, the proposed potentiostat is equipped with the ability to operate normally even if the battery is being charged. Care should be taken when dealing with power sharing, as it could pose a problem if not designed properly. In general, the rechargeable battery should not be drained and charged at the same time. It could damage the battery, or at least reduce its life span. An effective method to tackle this situation is to have a separate power

                                                                   

path for the load other than from the battery so that the process of charging the battery and supplying the load can occur simultaneously. This concept is called the load-sharing scheme. In our proposed design, we connected a pull-down resistor, a Schottky diode, and a low-drain-to-source resistance, high-threshold-voltage PMOS to act as the power-path controller, as illustrated in Fig 3.

## Analog Front-End (AFE)

Overall, the key components of the proposed potentiostat are a unipolar-to-bipolar converter, a current-to-voltage converter, a bipolar-to-unipolar converter, a low-pass filter, and an ADC input protection circuit, as shown in Fig 4. AFE utilizes several dedicated operational amplifiers (op amps), capacitors, and resistors that comprise different analog signal processing functions, which can be elaborated in detail in Fig 5.

**Unipolar-to-Bipolar Converter (UBC).** The DAC module can only output positive unipolar values, ranging from 0 V to 3.3 V (the supply voltage to the DAC module). Nevertheless, electrochemical analysis, particularly CV, requires not only a positive voltage sweep but also a negative one. Thus, it needs a block that can shift the incoming signal to cover the negative values as well. This circuit is called a unipolar-to-bipolar converter (UBC). UBC maps [0, 3.3] V input voltage to

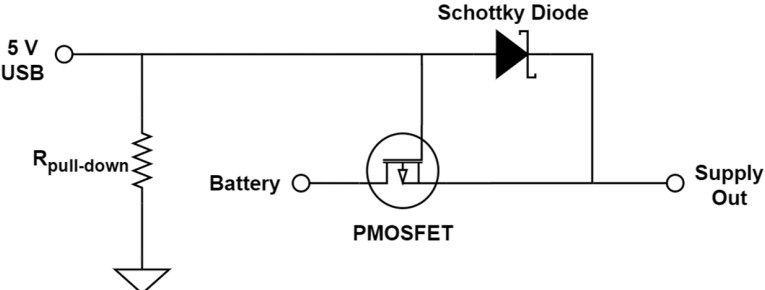

**Fig 3**. Power-path controller circuit.

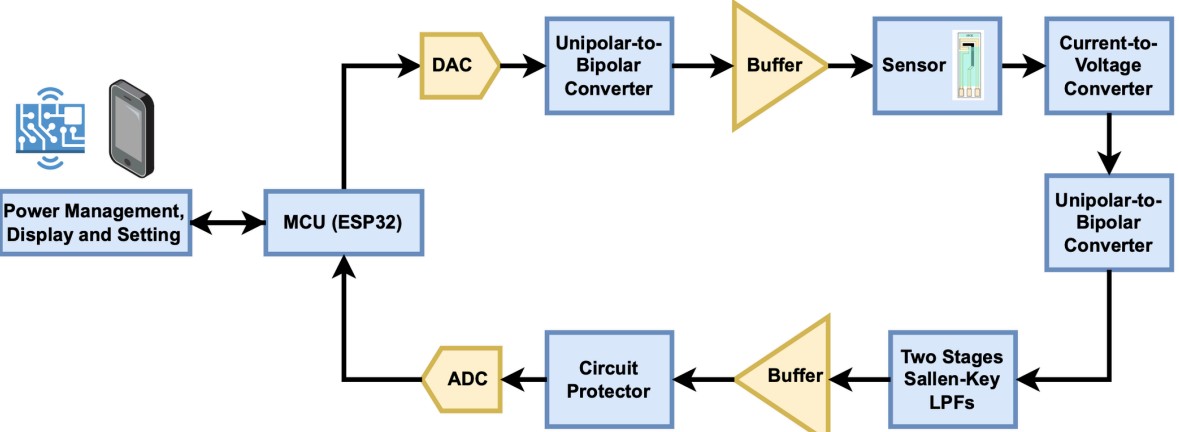

**Fig 4**. Block diagram of the proposed Analog Front-End (AFE) in GaneStat System.

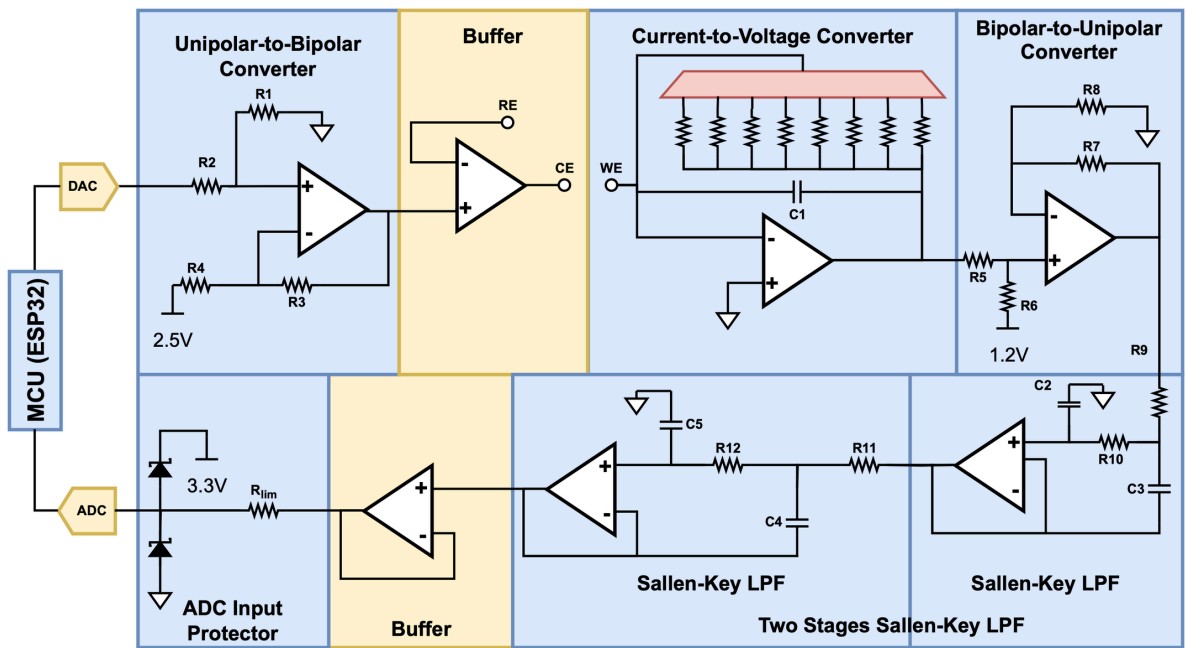

**Fig 5**. Detail of Analog Front-End (AFE) Schematic Diagram for GaneStat System.

[–2.5, 2.5] V output voltage according to the following linear function:

$$V_{UBC} = \left( \frac{V_{SWEEP,pp}}{V_{DAC,pp}} \right) \cdot V_{IN,UBC} + V_{OFS,1}, \tag{3}$$

where $V_{SWEEP,pp}$ is the desired voltage span, $V_{DAC,pp}$ is the peak-to-peak voltage of the DAC, $V_{IN,UBC}$ is the input voltage to UBC, and $V_{OFS,1}$ is the offset voltage. Substituting 3.3 V for $V_{IN,UBC}$ and 2.5 V for $V_{UBC}$ to Eq (3) gives the offset voltage of 2.5 V.

**Sensor buffer.** This buffer is employed to control the sweep potential provided to the reference cell's electrode. The topology of unity gain ensures zero current flow to the reference electrode, thereby enabling the voltage to remain stable as it serves as a reference for the working electrode. Additionally, this block is utilized to pre-empt loading effects resulting from the sensor's equivalent resistance and capacitance, which may disrupt the sweep voltage signal if the buffer is absent. It implies that the op-amp used has to have robust driving capability in addition to a low input bias current.

**Current-to-Voltage Converter (CVC).** Current will flow through the working electrode in response to the sweep voltage being applied. This current contains important information; however, the signal can only be processed by the microcontroller in the form of a voltage input. Therefore, the current input must be translated into voltage beforehand. This block is realized at the circuit level by a trans-impedance amplifier (TIA). In the DC state, the converted voltage is given by Eq (4) as follows:

$$V_{TIA} = I_{TIA} \cdot R_{gain}, \tag{4}$$

where $I_{TIA}$ is the current flowing through CVC and $R_{gain}$ is the feedback resistor. A compensation capacitor is introduced in the feedback path to prevent the output voltage from oscillating (which is caused by the capacitive load of the sensor), which is detrimental to the stability of the trans-impedance amplifier.

The maximum and minimum currents resulting from the electrochemical reaction depend on the sample being analyzed or the sensor's materials [25]. With this in mind, using a single feedback resistor would degrade the current reading resolution. As a result, only a few samples can be tested. To extend the current measurement range, the gain of the trans-impedance amplifier should be programmable. This implies that the feedback resistance can be modified to attain the appropriate reading resolution. One way to realize this is to have a programmable gain amplifier by adding an analog multiplexer and several resistors, where each resistance corresponds to a certain current range. This multiplexer acts as an electrical bridge that connects the inverting input of the op-amp with one of eight different feedback resistors. The operation is as follows: when the current flow is low, the resistor that is connected to the feedback line is a resistor with a high value, and vice versa. By selecting the appropriate resistor through the multiplexer, the transimpedance amplifier can be adjusted to accommodate different input currents with the corresponding resolutions, which improves the signal-to-noise ratio (SNR). This flexibility allows for accurate and customizable readings for a wide range of applications.

GaneStat employs ADG408, an 8-channel analog multiplexer, to provide eight options for the current range, starting from ±10 nA to ±10 mA. Each binary word is composed of three digital signals coming from the microcontroller as shown in Fig 6. Control over these signals depends on the selected current range. We can choose the necessary range through the software. In addition, the resistance is calculated not only based on the input current range but also on the output voltage of the trans-impedance amplifier, which is designed to span from −1.2 V to 2.1 V. The operational amplifier used as the transimpedance amplifier should have a wide dynamic range to accommodate a wide current range. Firstly, we need to choose an op amp that has robust noise performance since the potentiostat is commonly used to detect a small current (a few nanoamperes to several milliamperes). If the noise level is much higher than the signal level, the resolution will deteriorate. The lower the noise floor is, the higher the dynamic range will be. The OPA4197 was selected in this work as the transimpedance amplifier due to its low input noise voltage density and low input noise current density of 5.5 nV/$\sqrt{Hz}$ and 1.5 fA/$\sqrt{Hz}$ at 1 kHz, respectively.

Other important aspects, such as input bias current and input offset voltage, should also be taken into account, as they influence the accuracy of the measurement. The OPA4197 has a maximum input bias current of 20 pA and a 100 $\mu$V input offset voltage. These considerably low values allow a wide range of currents to flow, even with a substantial amount of resistance attached to the feedback path. As such, this can prevent the output voltage of the transimpedance amplifier

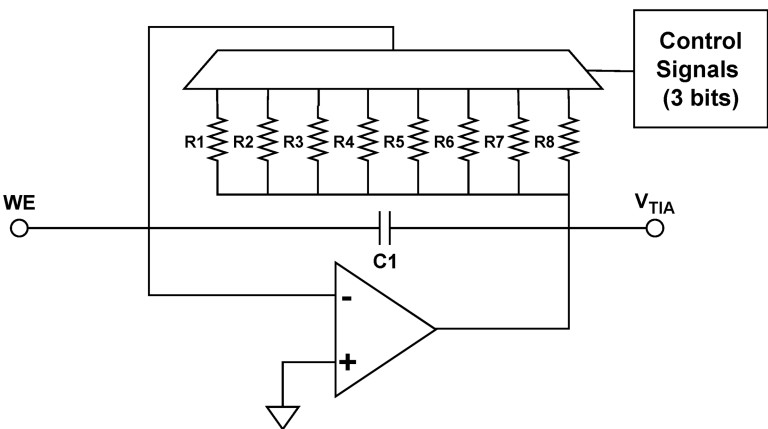

**Fig 6**. **Current-to-voltage converter circuit.**

from reaching saturation. The input bias current flows down the feedback resistor, resulting in an offset voltage as calculated in Equation (5):

$$V_{OB} = I_B \cdot R_{gain}. \tag{5}$$

The input offset voltage is also amplified but with a gain of only 1 V/V. Thus, the output offset voltage due to the input offset voltage is equal to 100 $\mu$V. The contributions of the DC offset at the output of the transimpedance amplifier from the input bias current ($I_B$) and input offset voltage ($V_{OS}$) when the current is zero (this corresponds to $V_{TIA} = 0V$) are shown in Table 1. The error ($\zeta$) is calculated using Equation (6) with respect to the ideal output voltage that is given in Equation (4):

$$\zeta = V_{OB} + V_{OS}. \tag{6}$$

It can be seen that the error produced when the input bias current and input offset voltage are present is sufficiently low, with a maximum value of only 0.3 mV. This is the advantage of utilizing the high noise performance of the OPA4197.

**Bipolar-to-Unipolar Converter (BUC).** The output voltage cannot be directly transmitted to the ADC because the ADC only permits positive unipolar voltage levels within the [0, 3.3] V range. However, the converted voltage signal has the potential to exhibit a negative value in instances where the current flow is negative, as per Eq (4). The level-converting concept is again adopted in this stage. This block translates the [–1.2, 2.1] V output voltage from the preceding stage to the [0, 3.3] V input voltage to the ADC according to Eq (7):

$$V_{BUC} = V_{TIA} + V_{OFS,2}. \tag{7}$$

Fig 7(a) depicts a difference amplifier to perform the level-converting process at the circuit level. By using Kirchoff's current law, we can express $V_{BUC}$ in terms of R1, R2, R3, and R4 as follows:

$$V_{BUC} = AV_{TIA} + BV_{OFF,2}, \tag{8}$$

with A and B are

$$A = \left( \frac{R_1}{R_3} \cdot \frac{R_3 + R_4}{R_1 + R_2} \right), \tag{9}$$

$$B = \left( \frac{R_2}{R_3} \cdot \frac{R_3 + R_4}{R_1 + R_2} \right). \tag{10}$$

Eq (7) implies that A and B must equal unity for both equations to maintain equivalency. Substituting these values into Eqs (9) and (10) and rearranging them yields $R_1 = R_2$ and $R_3 = R_4$. For simplicity, the values $R_1 = R_2 = R_3 = R_4 = R_{BUC}$ were selected.

**Table 1**. DC Offset Voltage Contributions from Input Bias Current ($I_B$) and Input Offset Voltage ($V_{OS}$) and the Calculated Error at Zero Input Current ($I_{TIA}$)

| Current Range ($\mu$A) | $V_{OB}$ (mV) | $V_{OS}$($\mu$V) | $\zeta$ (mV) |
|---|---|---|---|
| 10000 | 0.000002 | 100 | 0.100002 |
| 1000 | 0.0000165 | | 0.1000165 |
| 500 | 0.000036 | | 0.100036 |
| 100 | 0.0002 | | 0.1002 |
| 10 | 0.001774 | | 0.101774 |
| 1 | 0.01862 | | 0.11862 |
| 0.1 | 0.164 | | 0.264 |
| 0.01 | 0.3 | | 0.4 |

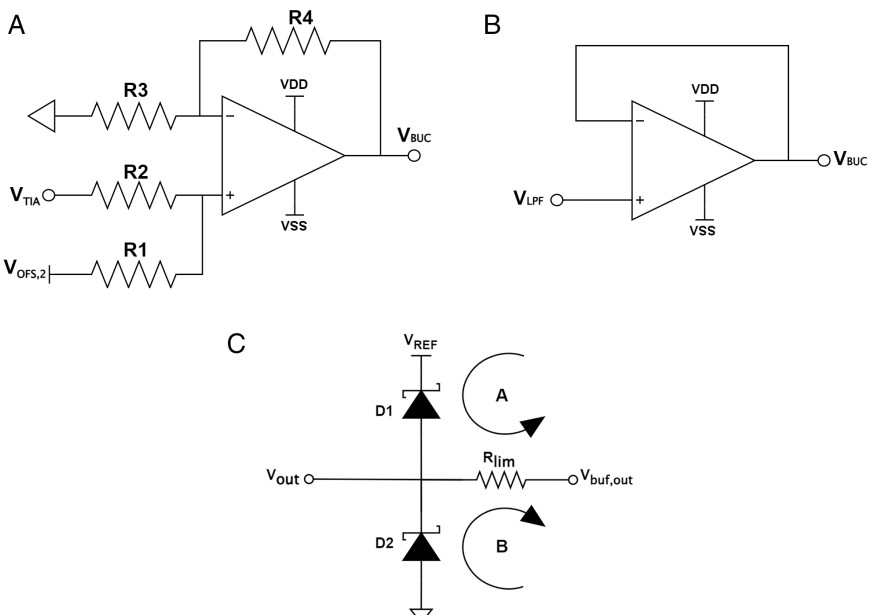

**Fig 7**. Circuits implementation of (a) Bipolar-to-Unipolar Converter, (b) Output Buffer, and (c) ADC Input Protection.

**Output buffer.** The reading accuracy might be corrupted if the impedance seen at the input of the ADC is too large [23]. To prevent this, a buffer was inserted before the ADC to improve the measurement [2], as the output impedance of a buffer is relatively small. The configuration of the buffer can be seen in Fig 7(b).

**ADC Input Protection Circuit (AIPC).** Although the use of a BUC block can keep the output voltage from exceeding the highest supply voltage (3.3 V) or even falling below the lowest permitted voltage (0 V), a surge current or voltage in transient conditions can still exist due to the overshoot. This condition is known as electrical overstress [26]. If it occurs, it will damage the ADC. One method to overcome this problem is by using an ADC input protection circuit (AIPC). The idea is to use two Schottky diodes connected in series to clamp the voltage entering the ADC input to a certain value if it lies outside the range of [0, 3.3] V. The anode of the top diode (D1) is attached to the same supply voltage as the ADC, while the other diode (D2) is connected to the ground, as illustrated in Fig 7(c). Having said that, this does not necessarily imply that the current passing through them can also be as big as possible, although Schottky diodes are capable of clamping. For that reason, a resistor is also used to limit the incoming current so that it does not damage both the diode and the ADC. A fast-switching, low-forward-voltage Schottky diode, BAT54, was chosen to perform the protection.

The procedure to obtain the appropriate resistance for the current-limiter resistor is as follows: assume that $V_{buf,out}$ has just exceeded the supply voltage to the ADC, $V_{REF}$. At this point, diode D1 will turn on and start to sink the current, whereas diode D2 will be cut off. The output voltage ($V_{out}$), then, will be subjected to the diode's behavior in such a way that the rate at which it increases begins to fall. The maximum value $V_{buf,out}$ can have is equal to the highest positive supply voltage to the op amp, which is $VDD$, whereas the maximum tolerable input voltage to the ADC is $V_{in,max}$. What we would want is that by the time $V_{buf,out}$ approaches $VDD$, $V_{out}$ should have been clamped to $V_{in,max}$ or lower. Thus, diode D1 needs to maintain a forward voltage, $V_{F,D1}$, of approximately $V_{in,max} - V_{REF}$. Let the diode's forward current, when the diode's forward voltage is $V_{F,D1}$, be $I_{F,D1}$. KVL on loop A gives the resistance value as expressed in Eq (11):

$$R_{lim,1} = \frac{VDD - V_{F,D1} - V_{REF}}{I_{F,D1}}. \tag{11}$$

By the same token, we can find the other possible value for the limiting resistor for the negative overstress voltage. Assume that $V_{buf,out}$ now goes below the ground. At this point, diode D2 will turn on and start to sink the current, whereas diode D1 will be cut off. As in the case of positive overstress voltage, the rate at which the output voltage decreases will be slower before being saturated to a certain value. This value is the minimum voltage that can be handled by the ADC, which is $V_{in,min}$ below the ground. This time, we would want $V_{out}$ to be clamped to-$V_{in,min}$ or higher when $V_{buf,out}$ reaches the minimum value it can be, which is $VSS$. Hence, diode D2 needs to maintain a forward voltage, $V_{F,D2}$, of approximately $-V_{in,min}$. The current flowing through D2 when its forward voltage equals $V_{F,D2}$ is $I_{F,D2}$. By using KVL on loop B, the other possible value for the current limiter resistor can be obtained as follows:

$$R_{lim,2} = \frac{VSS - V_{F,D2}}{I_{F,D2}}, \tag{12}$$

where the current limiter resistor, $R_lim$, can be obtained by finding the maximum between those calculated from Eqs (11) and (12), plus an extra margin.

Care should be taken when choosing the value at which $Vout$ will be clamped, as this affects how much forward voltage the diode should bear. This is also important, as a Schottky diode has a maximum forward voltage that is allowed at a certain forward current. In our case, the diodes will experience a forward voltage of around 0.3 V. In the worst-case condition, the forward current can be equal to the op amp's short circuit current, which is 65 mA. It is also known from the datasheet that the diode can tolerate up to 0.5 V and 0.8 V forward voltage at a forward current of 30 mA and 100 mA, respectively. Although the datasheet does not directly mention the forward voltage's tolerance at 65 mA forward current, we can safely infer that it lies somewhere between 0.5 V and 0.8 V. In other words, our choice of 0.3 V forward voltage will satisfy as well, even when the current that flows is the short-circuit current.

**Two-stage Sallen key low-pass filter.** Two second-order low-pass filters using Sallen Key LPFs are incorporated in the signal path to attenuate high-frequency noise resulting from previous stages. The overall cut-off frequency of the filters is set to 850 Hz with a 40 dB/decade roll-off. Sallen-key topology guarantees a unity gain in the low-frequency region. This is favorable, taking into account that the signal will no longer undergo any amplification, which in turn relaxes the calculation in the firmware.

Referring to the circuit in Fig 8, assuming that the potential between $R_1$ and $R_2$ is $V_1$, KCL analysis at the noninverting input of the op-amp gives:

$$\frac{V_{out}}{V_1} = \frac{\frac{1}{sC_2}}{R_2 + \frac{1}{sC_2}} = \frac{1}{1 + sR_2C_2}. \tag{13}$$

Performing KCL analysis at the junction between $C_1$, $R_1$, and $R_2$ yields:

$$\frac{V_1 - V_{in}}{R_1} + \frac{V_1 - V_{out}}{R_2} + \frac{V_1 - V_{out}}{1/sC_1} = 0. \tag{14}$$

Substituting Eq (13) to Eq (14) and proceeding with some algebraic simplification gives:

$$\frac{V_{out}}{V_{in}} = \frac{1}{1 + s(R_1C_2 + R_2C_2) + s^2R_1R_2C_1C_2}. \tag{15}$$

The Sallen-Key LPF used is a second-order LPF with a cutoff frequency of

$$f_c = \frac{1}{2\pi\sqrt{R_1R_2C_1C_2}}. \tag{16}$$

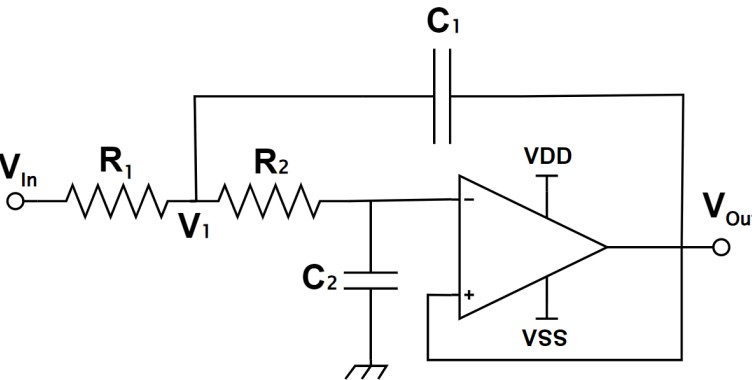

**Fig 8**. **Sallen-key low pass filter topology.**

The actual cutoff frequency obtained by the components is 861 Hz, with an 80 dB/decade roll-off with frequencies over the cutoff frequency. This is because the second-order Sallen-Key LPF is connected in cascade with another second-order Sallen-Key LPF, making the overall filter exhibit a fourth-order response. The attenuation is about 80 dB/dec, which is twice the specification mentioned before. This guarantees that the signal components that have a frequency of more than 861 Hz will be attenuated, which improves the system's performance in filtering noise components that have a high frequency so that the microcontroller can receive a clearer signal. Fig 9 compares the target frequency response with the system's achieved frequency response. It shows the simulated frequency response of the low-pass filter using MATLAB. A quick inspection of the graph gives a gain margin of 6 dB and a phase margin of 175 degrees. Both values are positive, which shows that the frequency response of the system is stable.

### PCB fabrication and printed case

Fig 10(a) shows the fabricated PCB along with the components that have been mounted onto it, resulting in an area of 6.29 cm × 6.22 cm. A 3D-printed case is used to protect the PCB from being exposed, as illustrated in Fig 10(b). The final implementation occupies a size of 6.95 cm × 6.85 cm × 3.26 cm, including the battery.

## Results

### System calibration

To adjust the accuracy of GaneStat, a dummy cell was used instead of a real solution. One can control how much current needs to flow to the working electrode by noting that only simple Ohm's law is involved in the calculation, as opposed to when calibrating the system using the prepared sample. The dummy cell consists of a single, fairly high-precision resistor, in which one end is attached to the counter and reference electrodes of the sensor. The other leg of the resistor is connected to the working electrode. The process of calibration can be deemed successful if the R-squared value of the regression line attains a value of unity and if the line traverses through the point of origin. Fig 11 shows the measurement of current flows when –0.6 V to 0.6 V potential is applied to the resistor with various values. Each resistor's value corresponds to a particular current range offered by GaneStat, as given in Table 2. The result infers that the curves already have good linearity, as indicated by the $R^2$ values. Furthermore, the vertical offset is considerably small that it can be neglected. Thus, the system has satisfied Ohm's law.

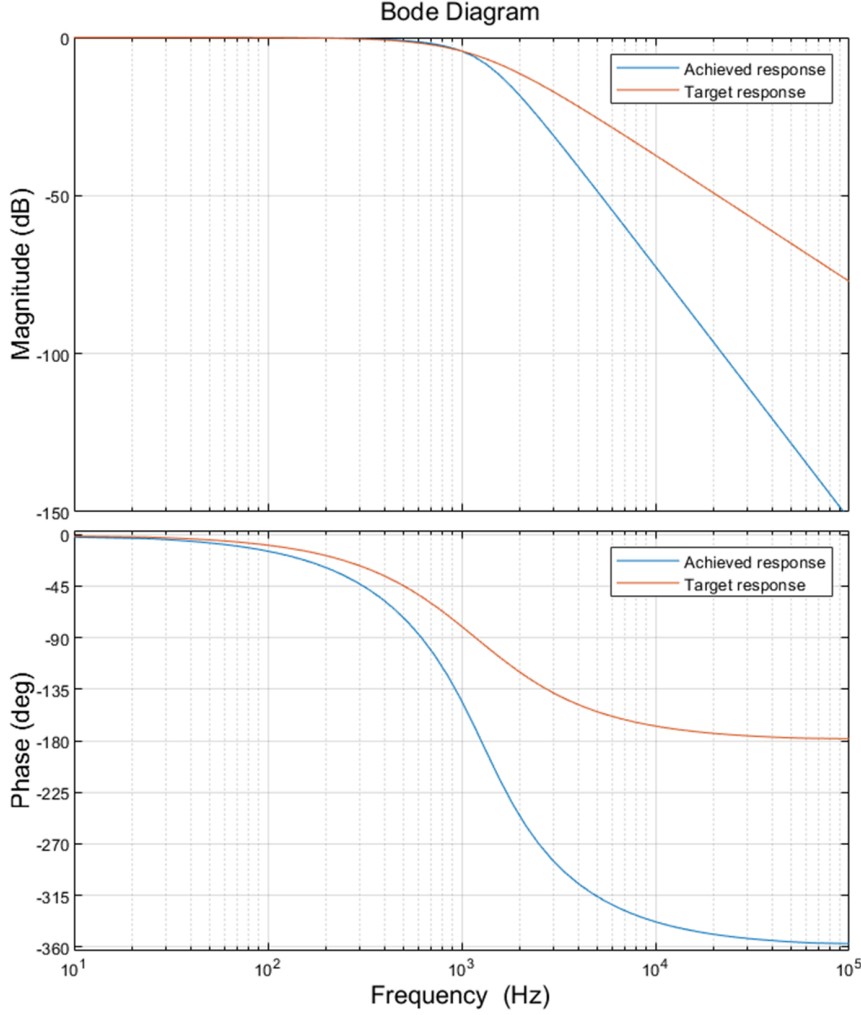

**Fig 9.** Bode plot of the two stages Sallen key LPF transfer function: Magnitude Response (top) and Phase Response (bottom).

## Performance comparison

As the potentiostat serves as a measuring device, possessing high accuracy is desirable to attain reliability. In addition, the quantity being measured in this paper is the Faradaic current's peak value. Accuracy can be obtained by knowing how large the difference between the measured value and the exact value is to get the error. The latter can be theoretically calculated by using the Randles-Sevcik equation [20]. It is also possible to validate the performance of the proposed potentiostat by using a commercial potentiostat and then compare the peak current readings drawn by both devices to get the relative accuracy. In this work, we utilized the EmStat4M module to perform the comparison.

The experiment was conducted by applying CV to 15 mM $K_3[Fe(CN)_6]$ solution using GaneStat with the test parameters as highlighted in Table 3. Next, the same procedure was conducted using EmStat4M. To highlight GaneStat's performance relative to EmStat4M, we used relative accuracy as a figure of merit, which is defined in Eq (17):

$$\eta_{pa(c)} = 100 - \left(\frac{|I_{pa(c),g} - I_{pa(c),e}|}{I_{pa(c),e}}) \times 100\right). \tag{17}$$

A

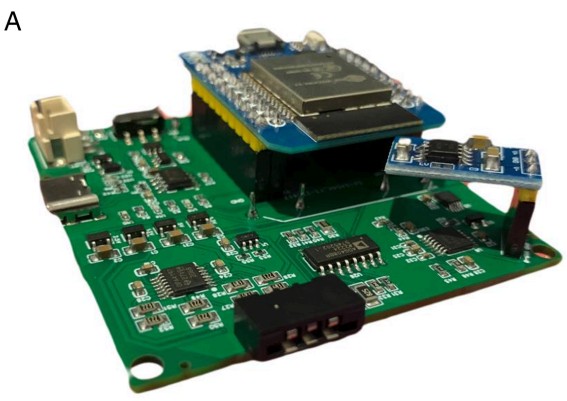

B

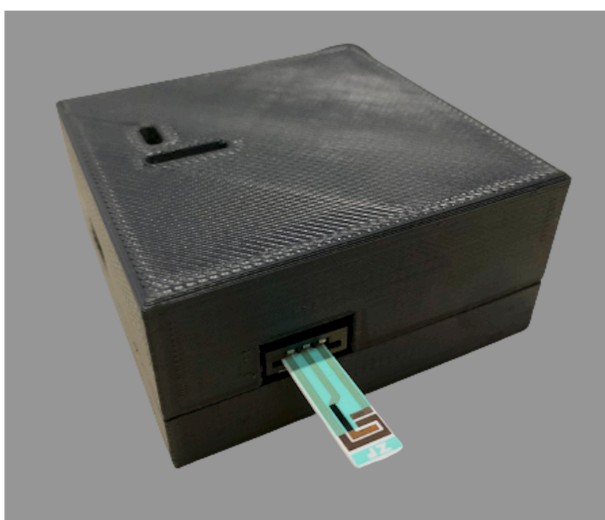

**Fig 10. Final implementation of GaneStat: (a) Fabricated PCB, (b) Printed case.**

Subsequently, we collected the anodic and cathodic peak currents obtained using GaneStat ($I_{pa,g}$ and $I_{pc,g}$) and EmStat 4 M ($I_{pa,e}$ and $I_{pc,e}$) from each cycle and plugged them into Equation (17) to get the relative accuracy as shown in Table 4.

It is worth noticing that the relative accuracy improved as the cycle length increased, peaking at 99.06% and 90.19% for anodic peak current and cathodic peak current, respectively. These results agree with [27], where it was demonstrated in the study that the bare SPE sensors exhibited an improvement after the umpteenth time of use. Furthermore, we can see from Fig 12 that there is only a slight difference in peak potentials between SPE tested in EmStat4M and SPE tested in GaneStat. However, we also found a higher difference in peaks in cathodic currents. We are considering that the quality of SPEs sometimes differs due to the manufacturing process of screen printing, where the surface is not smooth and not densely packed (in comparison to glassy carbon electrodes). Nevertheless, during our evaluation of the sample under varying concentrations from 2.5 mM to 20 mM, both SPEs (tested using EmStat and GaneStat) still maintained good linearity in both anodic and cathodic currents (Fig 12d). Thus, we conclude that the SPEs have no problem being tested for electrochemical analyses. Correspondingly, we can also conclude that the GaneStat has performed well in comparison to the commercial EmStat. The p-value obtained is 0.000124 which is less than 0.05, hence this shows that the current

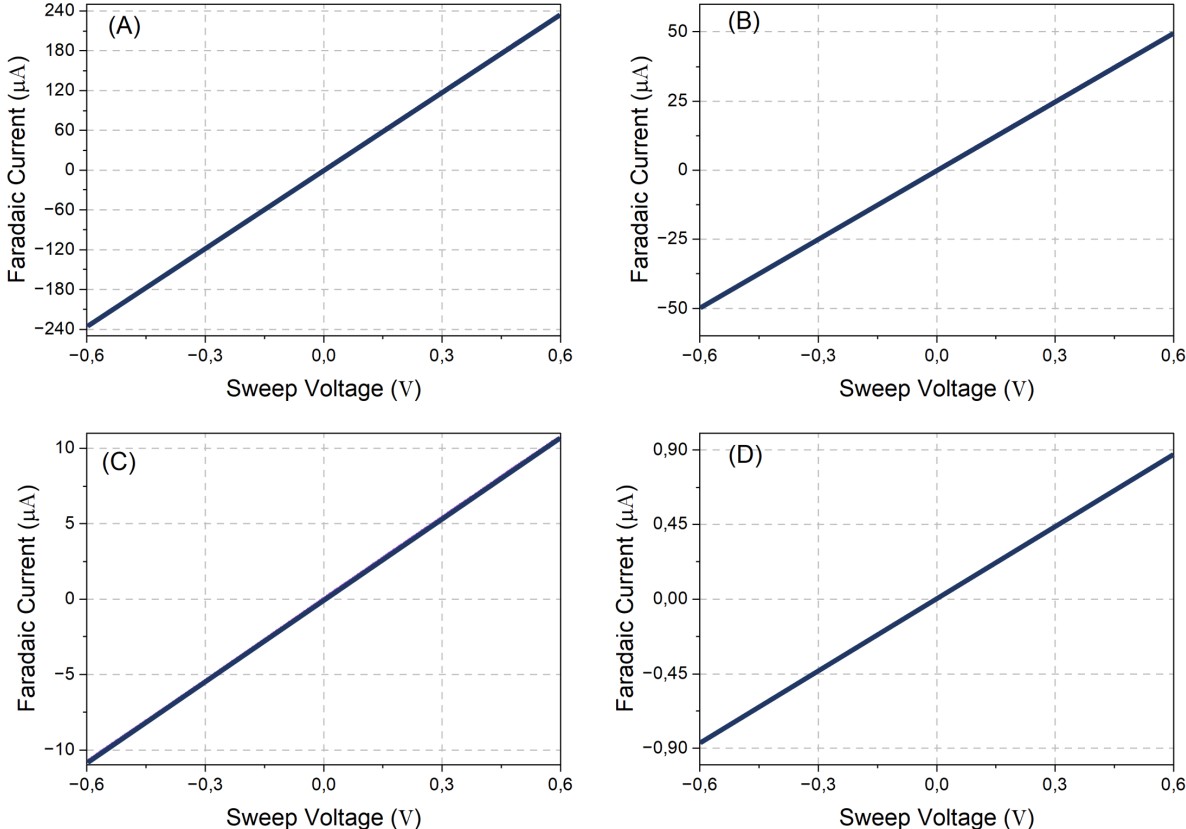

**Fig 11**. **Measurement calibration of GaneStat.**

**Table 2**. **Quantitative summary of system calibration.**

| Resistor ($k\Omega$) | Current Range ($\mu A$) | Regression Line | $R^2$ |
|---|---|---|---|
| 2.7 | 500 | $I = 393.32V - 0.2736$ | 0.9999 |
| 12 | 100 | $I = 83.031V - 0.0494$ | 0.9999 |
| 56 | 10 | $I = 18.01V - 0.0146$ | 0.9998 |
| 680 | 1 | $I = 1.4524V - 0.0017$ | 1 |

**Table 3**. **CV parameters for performance comparison.**

| Parameters | Value | Unit |
|---|---|---|
| Cycle | 3 | – |
| Scan Rate | 50 | mV/s |
| Step Size | 5 | mV |
| Initial Potential | −0.6 | V |
| Min. Potential | −0.6 | V |
| Max. Potential | 0.6 | V |
| Wait Time | 2 | s |

obtained is proportional to the concentration. The $R^2$ of the graph shows similar linearity between our off-chip GaneStat and reference on-chip EmStat4M.

**Table 4**. Performance comparison of GaneStat and EmStat4M based on CV response of 15 mM $K_3[Fe(CN)_6]$ solution.

| Cycle | 1 | 2 | 3 |
|---|---|---|---|
| **GaneStat Current ($\mu A$)** | | | |
| $I_{pa,g}$ | 102.42 | 97.91 | 96.85 |
| $I_{pc,g}$ | −93.48 | −92.51 | −91.69 |
| **EmStat4M Current ($\mu A$)** | | | |
| $I_{pa,e}$ | 84.26 | 93.44 | 95.95 |
| $I_{pc,e}$ | −107.83 | −103.14 | −101.67 |
| **Relative Accuracy (%)** | | | |
| $I_{pa}$ | 78.45 | 95.21 | 99.06 |
| $I_{pc}$ | 86.69 | 89.69 | 90.19 |

GaneStat was also able to perform other methods of electrochemical measurement, namely DIfferential Pulse Voltammetry (DPV), Linear Sweep Voltammetry (LSV), and Chronoamperometry (CA), as shown in Fig 13. The results are concluded in Tables 5, 6, and 7. The average error percentage for those methods respectively are 7.14%, 13.59%, and 6.28%. The test was done by performing three measurements for analytes of the same concentration, and taking the average peak current of each concentration. The result was then compared against the ones obtained from EmStat4 potentiostat.

## Electrical overstress protection verification and analysis

One of the features that GaneStat possesses is ADC input protection from the electrical overstress phenomenon. To verify this ability, we performed CV on a 2.7 kΩ resistor, and the result is shown in Fig 14. The curve can be divided into three regions: region I, region II, and region III. $V_{out}$ increases linearly with respect to $V_{sweep}$ in region II. As $V_{out}$ approaches 3.6 V, the rate of change begins to decline before saturated to slightly above 3.6 V. In the opposite direction, when $V_{out}$ falls below 0 V, the slope starts to decrease and eventually converges to approximately –0.3 V. It suggests that the protection system has worked properly.

The curve can be mathematically formulated to find the two points at which it transitions from the linear region to the logarithmic-behavior regions represented by regions I and II. First, substituting Eq (4) to Eq (7) and noting that $V_{sweep}$ is equal to the reference electrode's voltage yields a compact expression that relates $V_{sweep}$ and $V_{BUC}$ as follows:

$$V_{BUC} = \left(\frac{R_{gain}}{R_D}\right) V_{sweep} + V_{OFS,2}, \tag{18}$$

where $R_D$ is the nominal value of the dummy cell resistor. Note that the output voltage of the system, $V_{out}$, can be expressed as follows:

$$V_{out} = V_{BUC} - I_o R_{lim}, \tag{19}$$

where $I_o$ is the output current of the output buffer presented in Fig 7. It is worth noting that the above equation only holds when the signal frequency is sufficiently below the cut-off frequency of the low-pass filter block. Forcing the operation beyond this point will only attenuate the signal.

Next, combining Eqs (18) and (19) gives

$$V_{out} = \left(\frac{R_{gain}}{R_D}\right) V_{sweep} + V_{OFS,2} - I_o R_{lim}, \tag{20}$$

where this equation applies to all regions. Notice that the first two terms are linear with respect to $V_{sweep}$, whereas the last term exhibits nonlinear dependence on $V_{sweep}$ owing to the presence of $I_o$. As the curve stays in the linear region, both

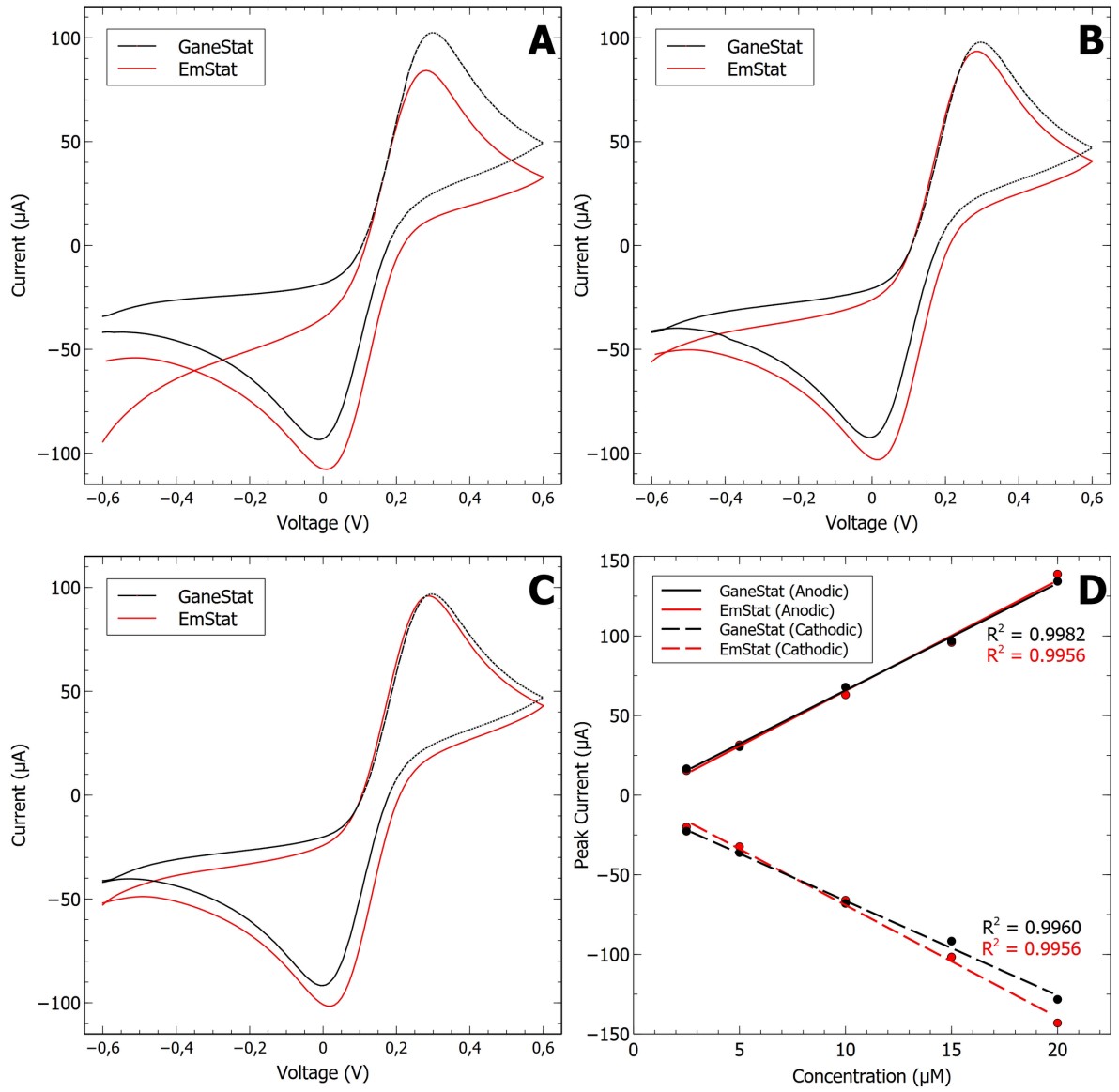

**Fig 12.** Comparison of FeCN's CV responses measured by GaneStat and EmStat4M at 15 mM concentration in (A) Cycle 1, (B) Cycle 2, (C) Cycle 3, and (D) Calibration curve for GaneStat and EmStat4M at concentration of 2.5, 5, 10, 15, and 25 mM.

Schottky diodes in Fig 7 are off. Therefore, only the output buffer op-amp contributes to $I_o$. Although $I_o$ is nonlinear, its effect in the linear region is not significant as the current-limiter resistance, $R_{lim}$, is small enough that the variation can be neglected. However, this is not the case in the other two regions since one of the diodes will conduct and force $I_o$ to obey Shockley's diode equation as indicated by

$$I_o = I_s \left( e^{\frac{V_F}{nU_T}} - 1 \right),$$

(21)

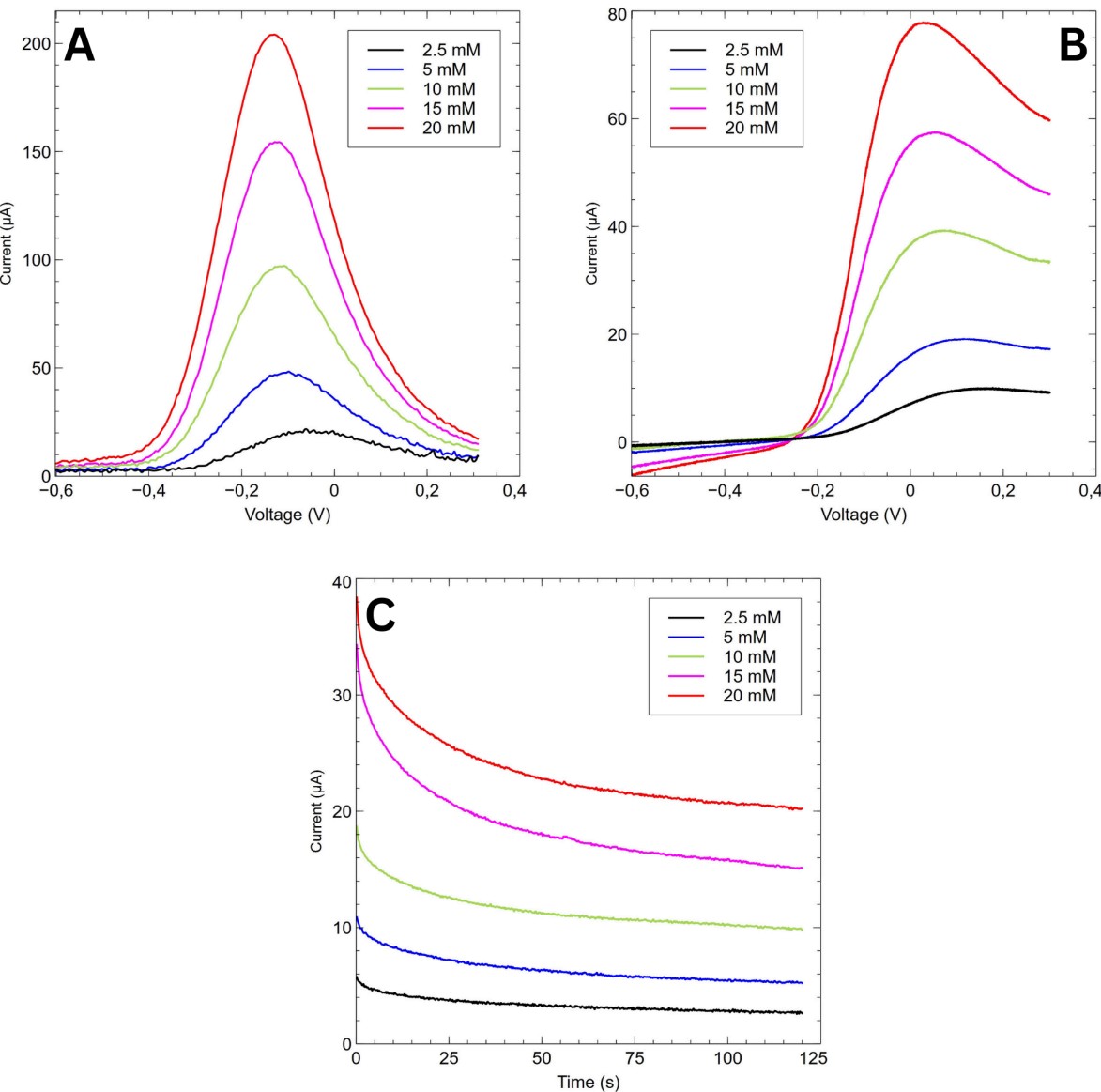

**Fig 13**. FeCN's electrochemical measurements of various methods performed by various methods: (A) DPV, (B) LSV, and (C) CA.

where $I_s$ is the scale current which depends on the geometry of the diode, $V_F$ is the diode's forward voltage, n is the diode's ideality factor, and $U_T$ is the thermal voltage. Putting this expression into Eq (20) gives Eq (22):

$$V_{out} = \left(\frac{R_{gain}}{R_D}\right) V_{sweep} + V_{OFS,2} - I_s R_{lim} \left(e^{\frac{V_F}{nU_T}} - 1\right). \tag{22}$$

We can also express $V_F$ in terms of $V_{out}$ and $V_{REF}$ as in the following equation:

$$V_F = V_{out} - V_{REF}, \tag{23}$$

**Table 5**. Differential pulse voltammetry peak current data.

| Concentration (mM) | Current ($\mu$A) | Ref. Current ($\mu$A) | %Error |
|---|---|---|---|
| 2.5 | 18.28 | 17.76 | 2.92 |
| 5 | 40.10 | 35.99 | 11.43 |
| 10 | 82.35 | 80.86 | 1.84 |
| 15 | 147.60 | 127.1 | 16.13 |
| 20 | 180.19 | 174.3 | 3.38 |

**Table 6**. Linear sweep voltammetry peak current data.

| Concentration (mM) | Current ($\mu$A) | Ref. Current ($\mu$A) | %Error |
|---|---|---|---|
| 2.5 | 7.65 | 5.67 | 35.06 |
| 5 | 16.61 | 15.25 | 8.92 |
| 10 | 34.61 | 30.97 | 11.77 |
| 15 | 52.15 | 48.78 | 6.92 |
| 20 | 71.42 | 67.83 | 5.29 |

**Table 7**. Chronoamperometry steady-state current results.

| Concentration (mM) | Current ($\mu$A) | Ref. Current ($\mu$A) | %Error |
|---|---|---|---|
| 2.5 | 2.22 | 2.02 | 10.23 |
| 5 | 4.22 | 3.95 | 6.82 |
| 10 | 8.70 | 8.55 | 1.70 |
| 15 | 13.78 | 12.59 | 9.48 |
| 20 | 18.55 | 19.16 | 3.19 |

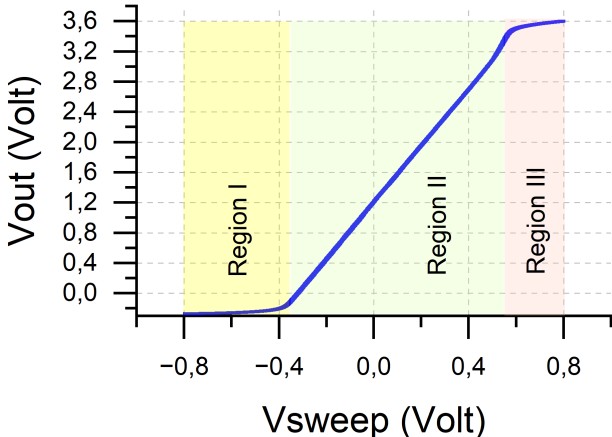

**Fig 14**. Voltage measured at the ADC input under electrical overstress condition.

where substituting this into Eq (22) and rearranging the terms involving $V_{out}$ on the left-hand side yields:

$$V_{out} + \alpha e^{\frac{V_{out}}{nU_T}} = \beta V_{sweep} + V_{OFS,2} + \gamma$$

$$\alpha = I_s R_{lim} e^{\frac{-V_{REF}}{nU_T}}$$

$$\beta = \frac{R_{gain}}{R_D}$$

$$\gamma = I_s R_{lim},$$

(24)

which assume that the variation due to $I_o$ can be neglected in the linear region. Thus, Eq (20) can be written as:

$$V_{out} \approx \beta V_{sweep} + V_{OFS,2}. \tag{25}$$

Following that, replacing $\beta V_{sweep}$ in Eq (24) with the one that is obtained from Eq (25) results in the output voltage at which the curve enters region III from region II, $V_{out_{II-III}}$ as follows:

$$V_{out} + \alpha e^{\frac{V_{out}}{nU_T}} = V_{out} - V_{OFS,2} + V_{OFS,2} + \gamma \Leftrightarrow$$

$$e^{\frac{V_{out}}{nU_T}} = \frac{\gamma}{\alpha} \Leftrightarrow V_{out} = nU_T \cdot \ln\left(\frac{\gamma}{\alpha}\right). \tag{26}$$

Substituting $I_s R_{lim}$ and $I_s R_{lim} e^{\frac{-V_{REF}}{nU_T}}$ for $\gamma$ and $\alpha$, respectively, into Eq (26) yields $V_{out_{II-III}}$ as shown in the following equation:

$$V_{out_{II-III}} = nU_T \cdot \ln\left(\frac{I_s R_{lim}}{I_s R_{lim} e^{\frac{-V_{REF}}{nU_T}}}\right) \Leftrightarrow V_{out_{II-III}} = nU_T \cdot \ln\left(e^{\frac{V_{REF}}{nU_T}}\right) \Leftrightarrow$$

$$V_{out_{II-III}} = nU_T \cdot \frac{V_{REF}}{nU_T} \Leftrightarrow V_{out_{II-III}} = V_{REF}. \tag{27}$$

Substituting Eq (27) back either into Eq (24) or Eq (25) results the corresponding sweep voltage, $V_{sweep_{II-III}}$ as follows:

$$V_{sweep_{II-III}} = \frac{V_{REF} - V_{OFS,2}}{\beta} = \frac{V_{REF} - V_{OFS,2}}{\frac{R_{gain}}{R_D}}. \tag{28}$$

The values of $V_{REF}$, $V_{OFS,2}$, $R_{gain}$, and $R_D$ used in this experiment are 3.3 V, 1.2 V, 10 kΩ, and 2.7 kΩ, respectively. Substituting these values into Eq (28) yields $V_{sweep_{II-III}} = 0.567\ V$ and $V_{out_{II-III}} = 3.3\ V$, as expected when carefully inspecting the graph in Fig 14 directly.

By the same token, we can find the other point on the curve at which the transition from Region I to Region II occurs. The difference is that now the only diode that turns on is diode D2. Take the negative sign of $I_o$ from Eq (21) and substitute it into Eq (20) to get the following:

$$V_{out} = \beta V_{sweep} + V_{OFS,2} + \gamma \left(e^{\frac{V_F}{nU_T}} - 1\right). \tag{29}$$

Since the anode of diode D2 in Fig 7 is attached to the ground, the forward voltage of this diode becomes:

$$V_F = 0 - V_{out} = -V_{out}. \tag{30}$$

Now, substituting this into Eq (29) gives the transcendental equation as in Eq (24), but with different diode's forward voltage as follows:

$$V_{out} - \gamma e^{\frac{-V_{out}}{nU_T}} = \beta V_{sweep} + V_{OFS,2} - \gamma. \tag{31}$$

By substituting $\beta V_{sweep}$ with the one that is in Eq (25) results the output voltage, $V_{out_{I-II}}$, where the transition takes place as follows:

$$V_{out} - \gamma e^{\frac{-V_{out}}{nU_T}} = V_{out} - V_{OFS,2} + V_{OFS,2} - \gamma \Leftrightarrow$$

$$e^{\frac{-V_{out}}{nU_T}} = 1 \Leftrightarrow V_{out_{I-II}} = 0 \; V. \tag{32}$$

As in the case of the first transition point, to get the corresponding sweep voltage, $V_{sweep_{II-III}}$, we substitute Eq (32) back into Eq (25). The result is as follows:

$$V_{sweep_{II-III}} = \frac{-V_{OFS,2}}{\beta} = \frac{-V_{OFS,2}}{\frac{R_{gain}}{R_{lim}}} = -0.324 \; V. \tag{33}$$

The output voltage and its associated sweep voltage are found in Eqs (32) and (33), which agree with the experiment result in Fig 14. For the sake of clarity, we present the piecewise function of the output voltage, $V_{out}$, that is valid across the regions as follows:

$$V_{out} = \begin{cases} \beta V_{sweep} + V_{OFS,2} + \gamma \left( e^{\frac{V_{F_1}}{nU_T}} - 1 \right) & , V_{sweep} < a \\ \beta V_{sweep} + V_{OFS,2} & , a \leq V_{sweep} \leq b \\ \beta V_{sweep} + V_{OFS,2} - \gamma \left( e^{\frac{V_{F_2}}{nU_T}} - 1 \right) & , V_{sweep} > b, \end{cases}$$

where $a = V_{sweep_{I-II}}$, $b = V_{sweep_{II-III}}$, $V_{F_1} = V_{out} - V_{REF}$, and $V_{F_2} = -V_{out}$.

## Noise and resolution analysis by SPICE simulation

To characterize the system's performance in terms of noise immunity, we simulated the AFE circuit using spice models of each component. The first stage will be the current-to-voltage converter (CVC), at which the current signal from the sensor is sensed, while the output will be taken at the ADC input (Fig 5, from the WE node (CVC) to the output of the ADC input protection circuit). For each feedback resistor ($R_{gain}$), the simulated input-referred noise current as well as the calculated dynamic range (DR) at zero input current are shown in Table 8. Note that the eight feedback resistors are depicted in Fig 5 at the CVC block that is controlled by a multiplexer.

**Table 8**. Simulated input-referred noise current and calculated Dynamic Range (DR) at the Working Electrode (WE) Pin of the Current-to-Voltage Converter (CVC) at Zero Input and a Signal Bandwidth of 1 kHz.

| $R_{gain}(k\Omega)$ | Current Range ($\mu A$) | $I_{n,rms}$ (nA) | DR (dB) |
|---|---|---|---|
| 0.1 | 10000 | 112.79 | 98.95 |
| 0.825 | 1000 | 13.68 | 97.28 |
| 1.8 | 500 | 6.29 | 98.01 |
| 10 | 100 | 1.25 | 98.09 |
| 88.7 | 10 | 0.54 | 85.29 |
| 931 | 1 | 0.53 | 65.54 |
| 8200 | 0.1 | 0.53 | 45.54 |
| 15000 | 0.01 | 0.53 | 25.54 |

The dynamic range is expressed mathematically as follows:

$$DR = 20 \cdot \log_{10}\left(\frac{I_{max}}{I_{n,rms}}\right), \tag{34}$$

where $I_{max}$ is the maximum current signal at the input of the current– to– voltage converter and $I_{n,rms}$ is the total RMS input noise current. Overall, the obtained dynamic range at each current range option is adequate for basic electrochemical experiments. Nevertheless, it is always recommended to select higher current range options rather than the lower ones to avoid signal loss due to higher noise power. The minimum detectable current by the current-to-voltage converter is determined by the noise itself [29,30]. However, since the current is sampled by the ADC at the final stage, the current resolution ($\Delta I$) is re-defined as the maximum between the figure calculated from the ADC formula as in Eq (35) and the total RMS input noise current obtained earlier as follows:

$$\Delta I = \frac{I_{TIA,p-p}}{2^N}. \tag{35}$$

It can be seen from Table 9 that our proposed potentiostat has achieved maximum and minimum current resolutions of 0.3 uA and 0.53 nA, respectively.

## Comparison to other methods

Our proposed potentiostat, GaneStat, was compared with the EmStat4M module and other potentiostats that are designed to aim for portability and use a battery or portable power supply for their main energy to start the system. To be more fair, we calculated the sweep voltage resolution ($\Delta V_{sweep}$) using the formula given in Eq (36). As for the current resolution, we decided to use Eq (35) for comparison as follows:

$$\Delta V_{sweep} = \frac{V_{sweep,p-p}}{2^M}, \tag{36}$$

where $V_{sweep,p-p}$ is the peak-to-peak sweep voltage and M is the number of DAC bits. We briefly summarized the comparison in Table 10.

ESPotensio [2] offers multichannel measurement so that it can conduct up to three different experiments in parallel, whereas GaneStat can handle only one experiment at a time. However, this makes ESPotensio's footprint larger than GaneStat's. Also, ESPotensio employs non-rechargeable batteries instead of rechargeable ones to power up the device, therefore leaving only the USB cable as the main supply when the batteries have been drained completely. ESPotensio can provide a sweep voltage resolution of 0.7 mV $\left(\frac{3}{2^{12}}\right)$, which is 0.5 mV better than GaneStat. Nevertheless, it can exert

Table 9. Current resolution of GaneStat for each current range option.

| $R_{gain}(k\Omega)$ | Current Range ($\mu A$) | $\Delta I(\mu A)$ |
|---|---|---|
| 0.1 | 10000 | 0.3 |
| 0.825 | 1000 | 0.031 |
| 1.8 | 500 | 0.015 |
| 10 | 100 | 0.00305 |
| 88.7 | 10 | 0.00054 |
| 931 | 1 | 0.00053 |
| 8200 | 0.1 | 0.00053 |
| 15000 | 0.01 | 0.00053 |

**Table 10**. Comparison between GaneStat and Other Works

| Reference | GaneStat | EmStat4M | ESPotensio | MiniStat | eSTAT | SStat | PLP |
|---|---|---|---|---|---|---|---|
| | (This Work) | [33] | [2] | [31] | [32] | [11] | [34] |
| Dimension | 6.95 cm × 6.85 cm × 3.26 cm | 6.2 cm × 4.0 cm × 0.7 cm | 10 cm × 9 cm × 5 cm | 2.7 cm × 2.0 cm | 14 cm × 6.8 cm × 5.2 cm | N/A | 11.5 cm × 7.5 cm × 3.0 cm |
| Weight | 83.1 $g$ | 30 $g$ | N/A | N/A | 260 $g$ | N/A | N/A |
| Power Supply | Battery, USB-C | USB-C | Battery, USB | Battery | Powerbank | Battery | Battery |
| Battery Management System | Yes Charging, Load Sharing | N/A | No | No | No | No | Yes Charging |
| Experimental Capabilities | CV, LSV, DPV, CA | CV, LSV, DPV, SWV, NPV, CA, MA, OCP, etc. | CV, LSV, DPV, SWV, NPV, CA | CV, LSV, ASV, SWV, CA | CV | CV, LSV SWV, CA | CV |
| Communication | Bluetooth Classic | USB-C | BLE, USB | USART | BLE | BLE, Wifi | Bluetooth Classic |
| DAC bits | 12 | 16 | 12 | 15 | 8 | 12 | 12 |
| ADC bits | 16 | 16 | 12 | 15 | 12 | 12 | 12 |
| Sweep Voltage Range | ±2.5 V | ±3 V | ±1.5 V | ±1.2 V | ±1.5 V | ±1.5 V | ±1.5 V |
| Sweep Voltage Resolution | 1.2 mV | 100 $\mu$V | 0.7 mV | 0.6 mV | 11.7 mV | 0.7 mV | 0.7 mV |
| Current Range | ±10 nA – ±10 mA | ±1 nA – ±10 mA | ±500$\mu$A | ±100$\mu$A | ±300$\mu$A | ±10$\mu$A – ±150$\mu$A | ±1$\mu$A – ±1 mA |
| Current Resolution | 0.53 nA – 0.3 $\mu$A | 30.52 fA – 0.3 $\mu$A | 0.24 $\mu$A | 6 nA | 0.16 $\mu$A | 4.88 nA – 73.24 nA | 100 nA – 1 mA |
| Max Dynamic Range (dB) | 98.95 | N/A | N/A | 70.37 | N/A | N/A | N/A |
| Overstress Protection | ADC Input Protection | N/A | No | No | No | No | No |
| Cost ($) (Form) | 98.55 (System) | On-Chip (System) | 21.4 (System) | 25 (Component) | 25.12 (System) | 50 (Component) | N/A (N/A) |

a maximum sweep voltage of only 1.5 V, which is 1 V smaller than what our design can do. The current measurement range of ESPotensio is also limited to ±500$\mu$A, and its current resolution is still much lower than the similar works in [31] and [32]. On the contrary, we have equipped our design with a higher ADC resolution and an 8-channel multiplexer, thus giving a finer reading and wider current measurement ability.

MiniStat [31] occupied an area of 2.7 cm by 2.0 cm, which is much smaller than that of GaneStat. However, this does not take into account the battery size or the external case. Apart from that, MiniStat is not equipped with a GUI, which is necessary for us to control the parameters for the analysis. MiniStat can obtain a lower voltage step size of 0.6 mV at the expense of the voltage span itself. GaneStat provides a peak-to-peak sweep voltage 2.6 V higher than that of MiniStat. MiniStat's current range is not adjustable, thus limiting its utilization only to analytes whose faradaic current lies within the given range. Conversely, GaneStat has several current range options that can be selected in the software, making it more versatile. At the same current range setting (±100 uA), GaneStat offers a wider dynamic range of 98.09 dB, yielding an almost 28 dB gap with that of MiniStat's. As the ESP32's internal DAC is only 8 bits, eSTAT [32] suffers from low sweep voltage resolution compared to other works. The current measurement range and current resolution of eSTAT are ±300$\mu$A and 0.16 $\mu$A, respectively. GaneStat appears to outperform eSTAT in the aforementioned aspects. The utilization

of through-hole components in eSTAT resulted in a significant increase in its overall size due to the space it occupied, not to mention the power bank required for the power supply.

SStat [11] achieved more than one option for the current reading range, starting from $\pm 10$ $\mu$A to $\pm$ 150 $\mu$A as the lowest and the highest, respectively. However, GaneStat's lowest and highest current range options are 1000$\times$ smaller and 66$\times$ greater than those of SStat, respectively. Also, the design presented in [11] used the ESP32, which is approximately 1.25$\times$ larger than the ESP32 D1 MINI and does not come with the battery or the case. Therefore, we expect that SStat is much larger and heavier than our design, even though Sarkar et al. did not report any information regarding its weight and size. PLP [34] has included a dedicated circuit for charging the battery. However, the device might experience a load-sharing problem as there is no path separator between the system and the battery. If the potentiostat is used while being charged and this situation is repeated multiple times, then it might cause damage to the battery. GaneStat, on the other hand, incorporates the power path controller circuit to solve this problem. Also, PLP is bulkier than GaneStat, where PLP consumes 66.72% more space than our design. Besides that, the presence of the overstress protection circuit makes GaneStat superior in terms of safety.

GaneStat offers the smallest form factor in terms of dimension, third only to EmStat4M and MiniStat. It is worth remembering that GaneStat's final dimension includes spaces for the external case and the battery, whereas the others solely include the PCB dimension. Besides that, GaneStat only weighs 83.1 g, including the case and the battery, which is a threefold reduction compared to eSTAT [32]. GaneStat provides two options to power it: either from a rechargeable battery or from a USB Type-C. GaneStat also accommodates the need for battery charging by employing a charging circuit that manages the battery's charging-discharging cycle. This circuit will continuously monitor the battery's voltage and compare it to a certain level. When the voltage reaches this value, the charging circuit will automatically terminate the charging process. In addition, GaneStat incorporates a power-path controller circuit to separate the charging path to the battery from the load, i.e., the potentiostat system, thus preserving the battery's health. Another novelty presented by GaneStat is the protection from overstress that can damage the ADC, which is not available in [2,11,31,32,34].

Although GaneStat is currently implemented on CV, DPV, LSV and CA experiments, it can be adjusted to conduct another type of electrochemical analysis based on voltammetry, such as NPV and SWV by modifying the firmware and the software without having to disassemble or redesign the hardware, based om our previous work in ESPotensio. Firmware from our previous work, ESPotensio [2], can be included and customized to the current GaneStat. This scalability, which is entailed by the modularity possessed by GaneStat, can shorten the development duration in case an additional type of analysis or specific user interface for certain analytes is to be added to the design, either for simplified version or comprehensive version [35,36].

Finally, even though GaneStat is more expensive than the proposed designs in [2,11,31,32], its superiority in terms of high performance, modularity, and circuit protection outweighs that disadvantage. We believe that the performance of this system in terms of voltage range, voltage resolution, current resolution, and current range will be beneficial for a wide range of biosensor and chemical sensor applications. Upscaling capabilities of the system into multi-channel and its flexibility in customizing the software could be potential for chemicals analyses that use screen-printed electrodes modified with active materials (label format approach) or biomolecular receptors (label-free format approach). In addition, due to the relatively low-cost bill of materials (less than 100 USD), deployment of such system will be highly possible for point-of-care testing. Current trends in wearable devices for health monitoring also open great future for wearable potentiostat targeting sweat contents analysis with some of them integrated to physical-signal responses [37–39].

## Conclusion

We have proposed a comprehensive design and modular analysis of portable potentiostat with low-cost, high-accuracy, battery-powered, and wide-current measurement for laboratory applications. We designed analog front end of the potentiostat with several modules such as unipolar-tobipolar converter (UBC), buffer, curent-to-voltage coverter (CVC),

bipolar-to-unipolar coverter (BUC), two-stage sallen key low-pass filter, and ADC input protection unit (AIPC). The potentiostat was calibrated by using several resistors to adjust the accuracy and showed good linearity, which is indicated by the $R^2$ value, and produces a negligible offset. The designed potentiostat is capable of conducting multi mode analysis, such as CV, DPV, LSV and CA experiments of a standard redox probe solution of ($K_3[Fe(CN)_6]$) for preliminary evaluation. GaneStat is designed to be optimal in portability as well as performance while maintaining a relatively low development cost of only $98.55 and robustness through overstress protection circuit that prevents the incoming voltage to the ADC from exceeding or falling below a certain level determined by the ADC.

## Author contributions

**Conceptualization:** Isa Anshori, Christian Reivan.

**Data curation:** Isa Anshori, Christian Reivan, Iqbal Fawwaz Ramadhan, Theodore Maximillan Jonathan, Rizky Indah Sari, Uperianti.

**Formal analysis:** Christian Reivan, Iqbal Fawwaz Ramadhan, Theodore Maximillan Jonathan, Rizky Indah Sari, Uperianti.

**Funding acquisition:** Isa Anshori, Infall Syafalni.

**Methodology:** Isa Anshori, Infall Syafalni.

**Supervision:** Infall Syafalni, Trio Adiono, Akhmadi Surawijaya.

**Visualization:** Iqbal Fawwaz Ramadhan, Theodore Maximillan Jonathan, Rizky Indah Sari.

**Writing – original draft:** Isa Anshori, Christian Reivan, Theodore Maximillan Jonathan.

**Writing – review & editing:** Infall Syafalni.

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
