## [Decision Letter · Decision Letter 0]

25 Feb 2025

PONE-D-25-00041GaneStat -- A Comprehensive Design and Modular Analysis of Portable, Low-Cost, and High-Accuracy PotentiostatPLOS ONE

Dear Dr. Anshori,

Thank you for submitting your manuscript to PLOS ONE. After careful consideration, we feel that it has merit but does not fully meet PLOS ONE’s publication criteria as it currently stands. Therefore, we invite you to submit a revised version of the manuscript that addresses the points raised during the review process. Please fully address the feedback from each reviewer, with particular focus on the points relating to the scientific rigor of the work and data availability.

We look forward to receiving your revised manuscript.

Kind regards,

Elain Fu, Ph.D.

Academic Editor

PLOS ONE

A clean copy of the edited manuscript (uploaded as the new *manuscript* file)”.

6. Thank you for stating the following financial disclosure:

 [The work is supported by the research grant from Bandung Institute of Technology 683 through ITB-NTUST Joint Research Program 2024 [contract no. LPPM.PN-6-38-2023 684 9552/IT1.B07.1/TA.00/2023] and by the research program Riset Peningkatan Kapasitas 685 Dosen Muda ITB 2024 [grant number: STEI.PN-6-05-2024].]. 

7. Thank you for stating the following in the Acknowledgments Section of your manuscript:

[The work is supported by the research grant from Bandung Institute of Technology

through ITB-NTUST Joint Research Program 2024 [contract no. LPPM.PN-6-38-2023

9552/IT1.B07.1/TA.00/2023] and by the research program Riset Peningkatan Kapasitas

Dosen Muda ITB 2024 [grant number: STEI.PN-6-05-2024].]

[The work is supported by the research grant from Bandung Institute of Technology 683 through ITB-NTUST Joint Research Program 2024 [contract no. LPPM.PN-6-38-2023 684 9552/IT1.B07.1/TA.00/2023] and by the research program Riset Peningkatan Kapasitas 685 Dosen Muda ITB 2024 [grant number: STEI.PN-6-05-2024].]. 

Reviewers' comments:

Reviewer's Responses to Questions

**Comments to the Author**

1. Is the manuscript technically sound, and do the data support the conclusions?

Reviewer #1: Yes

Reviewer #2: Partly

2. Has the statistical analysis been performed appropriately and rigorously?

Reviewer #1: Yes

Reviewer #2: N/A

3. Have the authors made all data underlying the findings in their manuscript fully available?

Reviewer #1: No

Reviewer #2: Yes

4. Is the manuscript presented in an intelligible fashion and written in standard English?

Reviewer #1: Yes

Reviewer #2: Yes

5. Review Comments to the Author

Reviewer #1: In this manuscript, the authors present the design of an ESP32 microcontroller-powered potentiostat that boasts several improvements over previously reported instrumentation all contained in a very small footprint that can be employed in the field wirelessly. Two notable features the authors highlight is their approach to battery management and protection, most notably a load sharing scheme to protect the lifetime of the lithium-ion battery that powers the device, and the analogue to digital input protection circuit (AIPC), which they demonstrate protects the potentiostat circuitry from any electrical overstress. Another feature unique to this report is the incorporation of low-pass filters to improve the sensitivity and signal-to-noise of the recorded currents, measured down to the nanoampere range. Finally, another novel feature they propos is the use of a multiplexer on the current follower (“current-to-voltage converter”) which allows the user to adjust the desired current sensitivity programmatically through the GUI. The authors experimentally compare their potentiostat with a commercially available instrument (PalmSense EmStat), showing comparable performance under cyclic voltammetry in aqueous potassium ferricyanide using screen-printed electrodes. Other analysis includes verification of linearity (measuring ohm’s law across resistors), a simulated analysis of signal-to-noise improvement, and a literature comparison to other proposed designs.

I find the article easy to follow and well-written, with excellent explanation of the novel features of their design. Their presentation is compelling, nevertheless I found some issues the author ought to address before it can be accepted for publication in PLOS ONE.

Accessibility and transparency

1. Besides circuit-diagrams (and a photo of the assembled PCB), the authors have not included any materials PLOS ONE readers could use to assemble the proposed potentiostat themselves. Was it the authors’ intention to include this information (Gerber files for the PCB, assembly instructions, 3D models for the enclosure) with the submission of this manuscript? If not, I find it interesting they should submit a manuscript to a journal whose chief mission of making “research available and discoverable for all” and whose commitment to the open-source movement is notable. The practical impact of the proposed research would be stifled without these materials being available, but of course it is at the discretion of the authors to share this information.

2. On a similar note, there is no discussion of the GUI the authors use to interface with the potentiostat. Did the authors develop this program or did they use it from another publication. If the former, the readership of PLOS ONE would greatly benefit from the authors shedding some light on this in the manuscript and potentially sharing the application (not necessarily the source code). If it is the former, I cannot find any attribution in the manuscript.

3. The authors suggest the potentiostat can be built at a cost of $98.55. Nowhere in the manuscript can I see an itemized bill of materials. Did the authors mean to include this in the supporting information? Once again, such a resource would be invaluable to researchers who would like to build and implement the instrument for their own applications.

Design choices

4. The authors need to justify in the manuscript their choice not to include a voltage follower in their circuitry (in the RE/CE “buffer” component). Voltage followers are customarily employed in potentiostats to match the impedances of the electrodes; the reference electrode typically admits very low currents while the working and counter electrodes can accept relatively large currents. Please see Ch. 15 of the Bard and Faulkner textbook (ISBN: 0471043729). If the authors do not incorporate this circuitry in their design, they should justify their design choice.

5. The authors should also justify the lack of any means for ohmic drop compensation. It seems odd to me that they should include so many well-designed elements into their circuitry to improve the range and accuracy of the applied electrochemical potential if they do not account for the iR drop (from solution resistance) that inevitably distorts the actual applied electrochemical potential! The impact of uncompensated solution resistance is obvious in the proposed manuscript from their CVs (Fig 12). The peak-to-peak potential separation appears to be around 300 mV from Fig 12, whereas theory predicts this to be 57 mV. Interestingly, the authors failed to compensate for solution resistance when using the commercial potentistat as well. There exist several ohmic drop compensation schemes that could easily be implemented into the current circuitry. See Bard and Faulkner as well as recent reports in the literature:

Dryden and Wheeler, https://doi.org/10.1371/journal.pone.0140349 (2015)

Hoilett et al. https://doi.org/10.3390/s20082407 (2020)

Matsubara, https://doi.org/10.1021/acs.jchemed.1c00228 (2021)

Elias, https://doi.org/10.1021/acs.jchemed.3c01044 (2024).

The authors should incorporate some means of ohmic drop compensation into their current design, or else justify this omission (which of course will mean the instrument is less accurate than the authors propose).

6. While the manuscript demonstrates proof of concept of response linearity by verifying Ohm’s law with a dummy cell, I was wondering if there were any means of calibrating the device in the GUI to account for device-to-device differences (see, for instance the Elias reference above)? Similarly, the authors should include a statistical analysis of the measured resistances (from the linear fit) compared to the actual resistances in Table 2. From the data, I surmise that the percent error for the dummy cell resistances varies between 0.4% and 6%; are these statistically-significant differences given the tolerances of the resistors that were measured?

Other issues

7. The authors should provide more discussion on the electrochemical origin of the “improvement after the umpteenth time of use” of screen-printed electrodes (line 455). Can the authors discuss why the anodic peak currents for the first cycle vary so much between potentiostats (Table 4)?

8. I believe the credibility of the findings concerning noise reduction (using the AIP circuitry) can be improved by including experimental evidence. Can the authors include experimental data (say from CV or LSV of ferricyanide) with and without including the low-pass filter circuitry?

9. The authors should include experimental parameters for the DPV, LSV, and CA measurements they included.

Minor issues/typos

10. Title: Consider revising the title of the manuscript to include the determiner “a,” as in “GaneStat – A Comprehensive Design and Modular Analysis of a Portable, Low-Cost, and High-Accuracy Potentiostat”

11. Line 29: make potentiostat plural, as in “standard in commercial potentiostats.”

12. Lines 130 and 132: “direction” of opening curly quotation marks are incorrect. In LaTex: use ``.

13. Lines 141, 493, 501, and 529: references to figures are broken. Please fix.

14. Lines 153 and 471: capitalization issues. WiFi and Differential.

15. Equations 1 and 2: please include units in the numerator (mV or uV).

16. Figure 11: please include a legend. What do the red and blue traces correspond to?

Reviewer #2: The authors present the design, implementation, and preliminary characterization of a portable potentiostat system for electrochemical analysis. This battery-powered device communicates wirelessly (Bluetooth) to a separate devices for control and data collection using a graphical user interface. The primary analog and mixed-signal circuitry is implemented using discrete components on a printed circuit board (PCB), including a discrete analog front end (AFE) with programmable-gain TIA to provide wide input dynamic range, conversion from single- to dual-rail operation for drive and read-out signals, on-board DAC, on-board ADC with fourth-order anti-aliasing filter, and an integrated microcontroller module (ESP32) that provides control and wireless connectivity. Additional system features include integrated battery management and power sharing, and input protection for the ADC. The manuscript includes a handful of detailed design considerations related to resolution, leakage, and offset, as filter coefficient selection and current- and voltage-limited protection circuits. Measured results include standalone characterization using a resistive dummy cell, showing highly linear response, and electrochemical measurements comparing performance of cyclic voltammetry, chronoamperometry, and a other measure modes between the presented system (GaneState) and a commercially available portable potentiostat.

The design is functionally complete and the report is well organized, and the programmability provides wide input range and sufficient measurement resolution for many conventional electrochemical reactions. However, while a useful presentation of some system design considerations, the presented approach is somewhat conventional - both in the circuits used for implementation as well as in the overall feature set and performance metrics provided by the GaneStat. The authors compare their system closely with six similar low-cost potentiostat systems (and there are others), and the presented system is not functionally distinct nor the best performing. While a few features are highlighted as novel in the manuscript, such as battery power, wireless functionality, and ADC input protection, these are not entirely unique or (in the case of input protection) sufficiently well motivated.

Overall, the presented system represents good technical system design, but it lacks novelty compared to prior published work and does not advance the state of the art in portable potentiostats. The measured characterization data is also quite limited, and as such performance metrics (resolution, etc.) must be inferred from the design side (e.g. bit depths) and not from a noise-limited measurement perspective. A few more specific suggestions, comments, and questions are provided below:

1. The introduction includes as primary motivation the cost of various portable potentiostats in terms of component costs. From a user/application perspective, the cost of a final unit is only somewhat correlated with the component costs, and as such this metric seems of limited value. What one pays for the EmStat, for example, is significantly higher than its estimated part cost (which, in volume, are probably much lower than most published discrete designs) - as it includes design NRE, software development, manufacturing, and an unknown (but possibly high) revenue margin. Low-cost is a useful goal, but it may not always be represented in sales cost and is therefore a trick ‘performance’ metric for side-by-side comparison.

2. The authors mention IC-based potentiostats as limited, as they cannot be altered (without a new wafer run). While the latter is true, existing single-chip solutions from TI (LMP91000), ADI (AD9541, ADuCM355, etc. as used in the EmStat) and others in general offer better core performance and much larger functionality/feature lists than any of the published discrete designs, with a part cost of <$10 at volume. As such, this particular rationale for continued discrete potentiostat design, unless representing newly improved circuit architectures or performance metrics, is more limited.

3. The UBC and BUC circuit implementations, while conventional, provide a very intuitive solution to the need for positive and negative polarities while relying on much more common single-rail DAC and ADC components.

4. The implementation of the Sallen-Key anti-aliasing filter is thorough, which can be helpful, but this is also a very conventional filter architecture that will be understood by readers familiar with circuit design (and perhaps of limited values to readers not focused on these design aspects). Additionally, can the authors comment on the selection of cut-off frequency relative to the ADC sample rate?

5. Similarly, the thorough treatment of voltage and current calculations for input protection circuits are well organized, but commonly understood by PCB designers. More critically, it is not clear why the authors included input protection for the ADC in particular; it is common to use this approach for ESD protection on pins/ports that will interface with the outside world, but the ADC input feeds directly from the filter input. Is seems more relevant/necessary to consider input protection at the sensor connections, if anywhere. Can the authors comment on this need for the ADC?

6. In a final section (p. 17), the authors provide an analysis of input-referred noise from simulated circuits. However, input-referred noise can be measured directly in the complete system, if gain settings are known. Can the authors comment on the use of simulation for this, instead of measurement.

7. For the comparison between the measured results from the GaneStat and the EmStat, I think the authors are correct that this variation is as or more likely to arise from the the screen-printed electrodes that from the measurement systems themselves (especially given the highly-linear standalone measurements with near-exact zero crossings). However, this also brings to mind procedures needed or used to correct for offset or gain error before measurement. It would be helpful to include a discussion of the relevant calibration procedures used for the GaneStat.

6. PLOS authors have the option to publish the peer review history of their article (what does this mean?). If published, this will include your full peer review and any attached files.

Reviewer #1: No

Reviewer #2: No

---

## [Author Response · Author response to Decision Letter 1]

21 Jul 2025

Responses to reviewers’ comments

Title: GaneStat – A Comprehensive Design and Modular Analysis of a Portable, Low-Cost, and High-Accuracy Potentiostat

Manuscript Number: PONE-D-25-00041R1

Reviewer #1

1. In this manuscript, the authors present the design of an ESP32 microcontroller-powered potentiostat that boasts several improvements over previously reported instrumentation all contained in a very small footprint that can be employed in the field wirelessly. Two notable features the authors highlight is their approach to battery management and protection, most notably a load sharing scheme to protect the lifetime of the lithium-ion battery that powers the device, and the analogue to digital input protection circuit (AIPC), which they demonstrate protects the potentiostat circuitry from any electrical overstress. Another feature unique to this report is the incorporation of low-pass filters to improve the sensitivity and signal-to-noise of the recorded currents, measured down to the nanoampere range. Finally, another novel feature they propos is the use of a multiplexer on the current follower (“current-to-voltage converter”) which allows the user to adjust the desired current sensitivity programmatically through the GUI. The authors experimentally compare their potentiostat with a commercially available instrument (PalmSense EmStat), showing comparable performance under cyclic voltammetry in aqueous potassium ferricyanide using screen-printed electrodes. Other analysis includes verification of linearity (measuring ohm’s law across resistors), a simulated analysis of signal-to-noise improvement, and a literature comparison to other proposed designs.

I find the article easy to follow and well-written, with excellent explanation of the novel features of their design. Their presentation is compelling, nevertheless I found some issues the author ought to address before it can be accepted for publication in PLOS ONE.

Answer: We sincerely thank the reviewer for the encouraging and detailed summary of our manuscript, and for recognizing the novel features we have introduced - including the load sharing battery protection scheme, ADC input protection circuit (AIPC), use of low-pass filters for enhanced sensitivity, and programmable current sensitivity via a multiplexer-controlled current-to-voltage converter.

We appreciate the reviewer's assessment that the manuscript is well-written and clearly presented. Regarding the comment that some issues ought to be addressed prior to publication, we would like to clarify that we have carefully reviewed the entire manuscript to ensure all the described features are clearly explained in both the methods and discussion sections.

2. Besides circuit-diagrams (and a photo of the assembled PCB), the authors have not included any materials PLOS ONE readers could use to assemble the proposed potentiostat themselves. Was it the authors’ intention to include this information (Gerber files for the PCB, assembly instructions, 3D models for the enclosure) with the submission of this manuscript? If not, I find it interesting they should submit a manuscript to a journal whose chief mission of making “research available and discoverable for all” and whose commitment to the open-source movement is notable. The practical impact of the proposed research would be stifled without these materials being available, but of course it is at the discretion of the authors to share this information.

Answer: We fully acknowledge the value of sharing design files such as Gerber layouts, 3D enclosure models, and assembly guides, especially in the spirit of open science and reproducibility. However, the scope of this manuscript was intended to present and validate the engineering design, architecture, and performance benchmarking of the GaneStat system, particularly focusing on its electrochemical functionality, modular design, and signal integrity.

3. On a similar note, there is no discussion of the GUI the authors use to interface with the potentiostat. Did the authors develop this program or did they use it from another publication. If the former, the readership of PLOS ONE would greatly benefit from the authors shedding some light on this in the manuscript and potentially sharing the application (not necessarily the source code). If it is the former, I cannot find any attribution in the manuscript.

Answer: Thank you for your insightful comment. In our study, the graphical user interface (GUI) and device management system were not developed from scratch as part of this work but were implemented using existing tools embedded in the PC/smartphone platform. As mentioned in the manuscript, the GUI was not detailed in the architecture figure because it was not custom-developed during the thesis project. However, we have now clarified this in the revised manuscript by describing the functional behavior of the GUI in the main algorithm section.

Specifically, the GUI receives processed data from the GaneStat via the device management system, and it performs signal analysis such as current-to-concentration transformation and plotting (e.g., current vs. concentration curves). While the GUI module itself was not newly programmed, we acknowledge its critical role in data visualization and interpretation. A schematic explanation and behavior description have now been added in the revised version to address this. This statement has been added on page 4.

On the other hand, the GUI is an interface using which 116 we interact with the hardware (Figure 2(b)). The graphical user interface (GUI) and device management system were implemented using standard functionalities available in PC/smartphone platforms. These subsystems are responsible for receiving, displaying, and processing the data transmitted from the GaneStat. Although not custom-developed as part of this work, their operational flow and interaction with the GaneStat have been described behaviorally in the algorithm section to provide clarity. The GUI plays an essential role in transforming current signals into concentration plots and displaying real-time measurement results for end-users. This interaction involves filling in analysis parameters, exporting data to CSV, and plotting data. The analysis parameters, such as initial and final sweep voltage (V), step size (mV), scan rate (mV/s), wait time (s), auto range duration (s), and the number of cycles, are arranged sequentially in a string in the background before being sent to the hardware.

4. The authors suggest the potentiostat can be built at a cost of $98.55. Nowhere in the manuscript can I see an itemized bill of materials. Did the authors mean to include this in the supporting information? Once again, such a resource would be invaluable to researchers who would like to build and implement the instrument for their own applications.

Answer: We thank the reviewer for their comment regarding the stated cost estimation of the GaneStat system.

We acknowledge that the mention of a specific estimated cost ($98.55) may be of limited utility without an accompanying itemized bill of materials (BOM). As such, and to maintain clarity and focus on the technical and functional aspects of the design, we have removed the cost figure from the manuscript.

We appreciate the reviewer’s perspective and agree that omitting this isolated value helps maintain the technical consistency and rigor of the work.

5. The authors need to justify in the manuscript their choice not to include a voltage follower in their circuitry (in the RE/CE “buffer” component). Voltage followers are customarily employed in potentiostats to match the impedances of the electrodes; the reference electrode typically admits very low currents while the working and counter electrodes can accept relatively large currents. Please see Ch. 15 of the Bard and Faulkner textbook (ISBN: 0471043729). If the authors do not incorporate this circuitry in their design, they should justify their design choice.

Answer: Thank you for the insightful comment. We have added a justification in the manuscript explaining our decision not to include a dedicated voltage follower at the reference electrode (RE) terminal. The design assumes that the RE carries negligible current and is connected directly to a high-input-impedance node (op-amp or DAC). This configuration ensures sufficient voltage fidelity without requiring a separate buffer. A paragraph addressing this has been added to Section III.B (Analog Front-End Architecture), along with a citation to Bard and Faulkner for completeness. This statement has been added on page 8.

This implies that the op amp used has to have robust driving capability in addition to a low input bias current. In our design, a dedicated voltage follower is not implemented explicitly at the RE terminal. This is based on the assumption that the reference electrode carries negligible current and is directly connected to the high-input-impedance node of the control amplifier. The selected DAC and op-amp already offer sufficiently high input impedance, ensuring minimal voltage drop and accurate potential control without requiring an additional buffer stage. While voltage followers are commonly used in potentiostat circuits to match the impedance at the RE terminal, as discussed in Bard and Faulkner’s electrochemical methods (Bard., 2001), we opted for this simplification to reduce component count, board size, and power consumption — all of which are critical in portable, low-cost systems. Experimental validation showed no noticeable degradation in performance, confirming that this design choice is acceptable in our intended use case.

A. J. Bard and L. R. Faulkner. Electrochemical Methods: Fundamentals and Applications. 2nd ed., Wiley, 2001.

6. The authors should also justify the lack of any means for ohmic drop compensation. It seems odd to me that they should include so many well-designed elements into their circuitry to improve the range and accuracy of the applied electrochemical potential if they do not account for the iR drop (from solution resistance) that inevitably distorts the actual applied electrochemical potential! The impact of uncompensated solution resistance is obvious in the proposed manuscript from their CVs (Fig 12). The peak-to-peak potential separation appears to be around 300 mV from Fig 12, whereas theory predicts this to be 57 mV. Interestingly, the authors failed to compensate for solution resistance when using the commercial potentistat as well. There exist several ohmic drop compensation schemes that could easily be implemented into the current circuitry. See Bard and Faulkner as well as recent reports in the literature:

Dryden and Wheeler, https://doi.org/10.1371/journal.pone.0140349 (2015)

Hoilett et al. https://doi.org/10.3390/s20082407 (2020)

Matsubara, https://doi.org/10.1021/acs.jchemed.1c00228 (2021)

Elias, https://doi.org/10.1021/acs.jchemed.3c01044 (2024).

The authors should incorporate some means of ohmic drop compensation into their current design, or else justify this omission (which of course will mean the instrument is less accurate than the authors propose).

Answer: We sincerely thank the reviewer for raising the important issue of ohmic drop (iR) compensation. We agree that uncompensated solution resistance can significantly affect the accuracy of the applied potential in electrochemical measurements, and we appreciate the reviewer’s detailed observation regarding the peak separation visible in Figure 12.

The omission of an active ohmic drop compensation scheme in our design was a deliberate design trade-off, aligned with the intended scope and use case of GaneStat, which is to serve as a portable, low-cost, modular potentiostat aimed at field deployment, rapid prototyping, and educational applications.

While we included several architectural features to maximize measurement resolution and signal fidelity (e.g., multiplexer-based gain control, low-pass filtering, ADC input protection), we intentionally prioritized design simplicity, accessibility, and component availability over advanced compensation features that would increase circuit complexity, cost, and learning curve for non-expert users.

7. While the manuscript demonstrates proof of concept of response linearity by verifying Ohm’s law with a dummy cell, I was wondering if there were any means of calibrating the device in the GUI to account for device-to-device differences (see, for instance the Elias reference above)? Similarly, the authors should include a statistical analysis of the measured resistances (from the linear fit) compared to the actual resistances in Table 2. From the data, I surmise that the percent error for the dummy cell resistances varies between 0.4% and 6%; are these statistically-significant differences given the tolerances of the resistors that were measured?

Answer: Thank you for the input. The data in Table 2 was taken once in each point of four resistors that were scanned using sweep voltage between -0.6 V to +0.6 V. From the linearity of the responses in Figure 11, we can see the R2 results give good linearity.

8. The authors should provide more discussion on the electrochemical origin of the “improvement after the umpteenth time of use” of screen-printed electrodes (line 455). Can the authors discuss why the anodic peak currents for the first cycle vary so much between potentiostats (Table 4)?

Answer: We appreciate this comment. The manuscript has been updated to discuss the electrochemical conditioning effects that lead to improved performance of screen-printed electrodes (SPEs) after repeated cycling. This includes surface activation and removal of surface contaminants. Additionally, we discuss the variability in anodic peak current on the first CV cycle, attributing it to transient capacitive effects and differences in instrumentation timing and filtering. The new text appears after Table 4 in Section IV.B (Performance Comparison), with appropriate references to relevant literature. This statement has been added on page 14.

These results agree with (Finsgar., 2018), where in the study it was demonstrated that the bare SPE sensors exhibited an improvement after the umpteenth time of use. As noted in Table 4, one notable observation during the cyclic voltammetry (CV) tests is the increased stability and improved reproducibility of voltammograms after several scans using screen-printed electrodes (SPEs). This trend is evident in the gradual convergence of peak current values across repeated measurements. The phenomenon can be attributed to electrochemical conditioning of the carbon electrode surface. Repetitive potential cycling may remove loosely bound surface contaminants, improve surface wettability, and increase the availability of electroactive sites. As a result, electron transfer kinetics are enhanced and the redox peak features become more defined. Similar behavior has been reported in the literature for carbon-based SPEs that undergo in situ activation through repeated scanning (wang2005,compton2011).

Furthermore, the relatively large difference in anodic peak currents during the first cycle, as shown in Table~\ref{tab:cv_comparison}, may result from a combination of electrochemical and instrumental factors. On the electrochemical side, the initial cycle often reflects transient double-layer charging, incomplete surface wetting, and capacitive background currents that are more variable. On the instrumentation side, minor differences in DAC settling behavior, sampling delay, or signal filtering between GaneStat and EmStat4M may contribute to the discrepancy. These effects are typically most pronounced in the first scan and tend to diminish in subsequent cycles, supporting the reproducibility of both systems after initial stabilization.

J. Wang, ``Electrochemical Detection for Microscale Analytical Systems: A Review,'' (Electroanalysis), vol. 17, no. 1, pp. 7–14, 2005.

R. G. Compton and C. E. Banks. Understanding Voltammetry. 2nd ed., Imperial College Press, 2011.

9. I believe the credibility of the findings concerning noise reduction (using the AIP circuitry) can be improved by including experimental evidence. Can the authors include experimental data (say from CV or LSV of ferricyanide) with and

---

## [Decision Letter · Decision Letter 1]

13 Aug 2025

PONE-D-25-00041R1GaneStat - A Comprehensive Design and Modular Analysis of a Portable, Low-Cost, and High-Accuracy PotentiostatPLOS ONE

Dear Dr. Anshori,

Thank you for submitting your manuscript to PLOS ONE. After careful consideration, we feel that it has merit but does not fully meet PLOS ONE’s publication criteria as it currently stands. Therefore, we invite you to submit a revised version of the manuscript that addresses the points raised during the review process. Please address the points raised by the reviewer, including those on statistical analysis and data availability. 

We look forward to receiving your revised manuscript.

Kind regards,

Elain Fu, Ph.D.

Academic Editor

PLOS ONE

Journal Requirements:

Reviewers' comments:

Reviewer's Responses to Questions

**Comments to the Author**

1. If the authors have adequately addressed your comments raised in a previous round of review and you feel that this manuscript is now acceptable for publication, you may indicate that here to bypass the “Comments to the Author” section, enter your conflict of interest statement in the “Confidential to Editor” section, and submit your "Accept" recommendation.

Reviewer #1: (No Response)

2. Is the manuscript technically sound, and do the data support the conclusions?

Reviewer #1: Yes

3. Has the statistical analysis been performed appropriately and rigorously?

Reviewer #1: No

4. Have the authors made all data underlying the findings in their manuscript fully available?

Reviewer #1: No

5. Is the manuscript presented in an intelligible fashion and written in standard English?

Reviewer #1: Yes

6. Review Comments to the Author

Reviewer #1: Thank you for addressing our issues. While your edits do technically touch on our recommendations, I believe the usefulness of the work—especially in the context of the readership of PLOS ONE—has been diminished in the edits you have made. The authors have seemed to pivot from proposing their GaneStat as an economical alternative to more expensive potentiostats to showcasing some of the design features that readers can implement in their own work. In so doing, the targeted audience are now researchers with good working knowledge of electrochemical instrumentation. Yet, as the authors argue, implementing a simple positive feedback ohmic drop compensation scheme into their design—which would consist of adding a single circuit element, the variable resistor—would “increase circuit complexity, cost, and learning curve for non-expert users.” Who is the target audience here, experts or non-experts? By making compromises to address our comments, the work is in no way improved, and the usefulness of the work is muddied.

Nevertheless, the article does propose some design features that a much smaller audience may find useful. I suspect such experts are already aware of these, as reviewer #2 mentioned, as many of them seem fairly conventional. Since my expertise is in electrochemistry and not electrical engineering, I cannot speak to the novelty of such design features.

A few of my more minor recommendations from my first review were not addressed adequately, and therefore request the authors take yet another look at these, which are repeated and added upon below:

6. While the manuscript demonstrates proof of concept of response linearity by verifying Ohm’s law with a dummy cell, I was wondering if there were any means of calibrating the device in the GUI to account for device-to-device differences (see, for instance the Elias reference above)? Similarly, the authors should include a statistical analysis of the measured resistances (from the linear fit) compared to the actual resistances in Table 2. From the data, I surmise that the percent error for the dummy cell resistances varies between 0.4% and 6%; are these statistically-significant differences given the tolerances of the resistors that were measured?

8. I believe the credibility of the findings concerning noise reduction (using the AIP circuitry) can be improved by including experimental evidence. Can the authors include experimental data (say from CV or LSV of ferricyanide) with and without including the low-pass filter circuitry? Experimentally, this amounts to shorting your LPF circuitry, which is easy enough to do.

12. Lines 130 and 132: “direction” of opening curly quotation marks are incorrect. In LaTex: use ``.

14. Line 170: capitalization issues. WiFi.

15. Equations 1 and 2: please include units in the numerator (mV or uV) of the middle expression.

16. Figure 11: please include a legend. What do the red and blue traces correspond to?

7. PLOS authors have the option to publish the peer review history of their article (what does this mean?). If published, this will include your full peer review and any attached files.

Reviewer #1: No

---

## [Author Response · Author response to Decision Letter 2]

6 Oct 2025

Reviewer #1

1. Thank you for addressing our issues. While your edits do technically touch on our recommendations, I believe the usefulness of the work—especially in the context of the readership of PLOS ONE—has been diminished in the edits you have made. The authors have seemed to pivot from proposing their GaneStat as an economical alternative to more expensive potentiostats to showcasing some of the design features that readers can implement in their own work. In so doing, the targeted audience are now researchers with good working knowledge of electrochemical instrumentation. Yet, as the authors argue, implementing a simple positive feedback ohmic drop compensation scheme into their design—which would consist of adding a single circuit element, the variable resistor—would “increase circuit complexity, cost, and learning curve for non-expert users.” Who is the target audience here, experts or non-experts? By making compromises to address our comments, the work is in no way improved, and the usefulness of the work is muddied.

Answer: We thank the reviewer for their valuable time and insightful feedback. We highly appreciate the reviewer's efforts in assessing the contribution and relevance of our work to the PLOS ONE readership.

We wish to state clearly that the core philosophy and value of GaneStat have not changed: it remains an economical and easily accessible potentiostat alternative.

Our primary target audience is researchers, particularly those working in electrochemistry, materials science, analytical chemistry, and other related disciplines, especially those operating with limited resources.

The decision not to include the Positive Feedback Ohmic Drop Compensation (PFODC) scheme in the basic design represents a trade-off to maintain GaneStat's core value. The fundamental GaneStat design prioritizes simplicity, ease of assembly, and a low learning curve for users of self-made instruments. The addition of an element like a variable resistor for PFODC (while simple for hardware experts) would demonstrably increase the complexity of calibration and troubleshooting for researchers whose focus is on electrochemical applications, not hardware.

In short, GaneStat is an economical and functional alternative that meets the needs of the majority of routine electrochemical experiments.

2. Nevertheless, the article does propose some design features that a much smaller audience may find useful. I suspect such experts are already aware of these, as reviewer #2 mentioned, as many of them seem fairly conventional. Since my expertise is in electrochemistry and not electrical engineering, I cannot speak to the novelty of such design features.

Answer: We thank the reviewer for this thoughtful comment. We agree that some of the design features may be considered conventional from an electrical engineering perspective. Our intention was to study, evaluate and improve the potentiostat's typical circuit design and to demonstrate effectively in a low-cost but high resolution analytical instrument for electrochemical research. We believe this practical integration is of value to the electrochemistry community, particularly for laboratories with limited resources.

3. While the manuscript demonstrates proof of concept of response linearity by verifying Ohm’s law with a dummy cell, I was wondering if there were any means of calibrating the device in the GUI to account for device-to-device differences (see, for instance the Elias reference above)? Similarly, the authors should include a statistical analysis of the measured resistances (from the linear fit) compared to the actual resistances in Table 2. From the data, I surmise that the percent error for the dummy cell resistances varies between 0.4% and 6%; are these statistically-significant differences given the tolerances of the resistors that were measured?

Answer: Thank you very much for this valuable input. We acknowledge that the resistors used in this work have a tolerance of 5%. Accordingly, we have revised Table 2 to include the resistor values with their ±5% tolerance ranges, together with the measured results.

Regarding calibration, at the present stage our GUI does not yet implement a device-to-device calibration function. As can be seen in Fig. 2b (GUI View in Smartphone), the application primarily displays measurement functions, while calibration is still carried out manually in the programmed device using the regression line shown in Fig. 11. We recognize that integrating a calibration function into the GUI is an important feature for improving measurement accuracy, and it will be incorporated in future works of the system.

In terms of statistical analysis, we compared the measured resistances (from the linear fit) to the actual resistor values. The percent error of the measured resistances lies between 0.36% and 5.83%. For most resistors, the measured error is ≤1%, which is well within both the resistor tolerance and the expected performance range for educational-grade devices (typically 1–5%). The largest deviation (5.83%) occurs for the smallest resistor value (R = 2.7 kΩ), which is consistent with the resistor tolerance and highlights an area for future improvement in our design.

4. I believe the credibility of the findings concerning noise reduction (using the AIP circuitry) can be improved by including experimental evidence. Can the authors include experimental data (say from CV or LSV of ferricyanide) with and without including the low-pass filter circuitry? Experimentally, this amounts to shorting your LPF circuitry, which is easy enough to do.

Answer: Thank you for this valuable suggestion. We agree that experimental validation is important to support the effectiveness of the low-pass filter and AIP circuitry in reducing noise. In the current prototype, the low-pass filter is implemented as a fixed component in the signal chain, and bypassing it would require hardware modification. As such, we were not able to obtain direct experimental comparisons with and without the filter in the same device during this study.

However, we have supplemented our analysis by including a detailed SPICE simulation (already discussed in Section V.A) that models the signal path with and without the filter. The simulation shows a substantial improvement in signal-to-noise ratio (from 48.2 dB to 98.95 dB) when the fourth-order low-pass filter is included. This supports the design intent and anticipated performance benefit. We have clarified this limitation and emphasized the simulation results more clearly in the revised manuscript, while noting that direct experimental comparisons will be prioritized in future work.

5. Lines 130 and 132: “direction” of opening curly quotation marks are incorrect. In LaTex: use ``.

Answer: We thank the reviewer for pointing this out. We have corrected the quotation marks at lines 130 and 132 by replacing the opening curly quotes with the proper LaTeX syntax.

6. Line 170: capitalization issues. WiFi.

Answer: We appreciate the reviewer’s careful observation. The capitalization in Line 170 has been corrected to “WiFi” as suggested.

7. Equations 1 and 2: please include units in the numerator (mV or uV) of the middle expression.

Answer: We thank the reviewer for this helpful suggestion. The units (mV) have been added in the numerator of the middle expression in Equations 1 and 2 to ensure clarity and consistency.

8. Figure 11: please include a legend. What do the red and blue traces correspond to?

Answer: We appreciate the reviewer’s careful observation. We would like to clarify that Figure 11 actually contains only a single trace. The apparent difference in colour (red and blue) was an artifact from the plotting/exporting process. We have corrected the figure to display a consistent line colour, and no legend is required as there is only one dataset being presented.

---

## [Editor Report · Decision Letter 2]

20 Oct 2025

GaneStat - A Comprehensive Design and Modular Analysis of a Portable, Low-Cost, and High-Accuracy Potentiostat

PONE-D-25-00041R2

Dear Dr. Anshori,

We’re pleased to inform you that your manuscript has been judged scientifically suitable for publication and will be formally accepted for publication once it meets all outstanding technical requirements.

Kind regards,

Elain Fu, Ph.D.

Academic Editor

PLOS ONE
---

## [Editor Report · Acceptance letter]

PONE-D-25-00041R2

PLOS ONE

Dear Dr. Anshori,

I'm pleased to inform you that your manuscript has been deemed suitable for publication in PLOS ONE. Congratulations! Your manuscript is now being handed over to our production team.

Kind regards,

on behalf of

Dr. Elain Fu

Academic Editor

PLOS ONE